# communications
# earth & environment

# Past rapid warmings as a constraint on greenhouse-gas climate feedbacks

Mengmeng Liu [1✉], Iain Colin Prentice [1,2,3], Laurie Menviel [4] & Sandy P. Harrison [3,5]

There are large uncertainties in the estimation of greenhouse-gas climate feedback. Recent observations do not provide strong constraints because they are short and complicated by human interventions, while model-based estimates differ considerably. Rapid climate changes during the last glacial period (Dansgaard-Oeschger events), observed near-globally, were comparable in both rate and magnitude to current and projected 21st century climate warming and therefore provide a relevant constraint on feedback strength. Here we use these events to quantify the centennial-scale feedback strength of $CO_2$, $CH_4$ and $N_2O$ by relating global mean temperature changes, simulated by an appropriately forced low-resolution climate model, to the radiative forcing of these greenhouse gases derived from their concentration changes in ice-core records. We derive feedback estimates (95% CI) of $0.155 \pm 0.035\,W\,m^{-2}\,K^{-1}$ for $CO_2$, $0.114 \pm 0.013\,W\,m^{-2}\,K^{-1}$ for $CH_4$ and $0.106 \pm 0.026\,W\,m^{-2}\,K^{-1}$ for $N_2O$. This indicates that much lower or higher estimates, particularly some previously published values for $CO_2$, are unrealistic.

[1] Department of Life Sciences, Imperial College London, London, UK. [2] Department of Biological Sciences, Macquarie University, North Ryde, NSW, Australia. [3] Ministry of Education Key Laboratory for Earth System Modelling, Department of Earth System Science, Tsinghua University, Beijing, China. [4] Climate Change Research Centre/ESSRC, The University of New South Wales, Sydney, NSW, Australia. [5] Department of Geography and Environmental Science, University of Reading, Reading, UK. ✉email: m.liu18@imperial.ac.uk

Climate warming leads to environmental changes with consequent feedbacks on climate[1,2]. Feedbacks involving the biosphere are generally positive owing to the nonlinear stimulation of all biological processes by increasing temperature[1,3]. However, the magnitude of biosphere feedbacks on centennial timescales relevant to current global warming is poorly known[3–6]. Estimates of the strength of individual feedbacks based on modern observations (e.g. ref. [7]) are hampered by the short length of the available records and uncertainties due to the influence of anthropogenic land-use change in recent decades. Earth System Models have been used to estimate the feedback strength[8–11], but many biosphere processes are either not included or are poorly represented in the current generations of models[12]. Indeed, even when biosphere feedbacks are included, these modules are often not used in future projections or in simulations of the past.

Dansgaard-Oeschger (D-O) events are rapid climate fluctuations that occurred about 25 times during the last glacial period (ca 115 to 11.7 ka). They are characterised by a rapid warming over a few decades followed by a slower cooling over centuries to millennia[13,14], with individual events registering warming of between 5 and 16 °C in Greenland[15]. This pattern is generally thought to reflect changes in the strength of the Atlantic Meridional Overturning Circulation (AMOC), whereby there is less poleward ocean heat transport when the AMOC is weak leading to cooling conditions around Greenland and vice versa[16,17]. The rapid warming events correspond to recovery of the AMOC. The cause of these events is still under debate and several mechanisms have been invoked, including ice-sheet instability[18], sea-ice fluctuations linked to ice-shelf growth and decay[19,20], sea-ice variability[21,22], shifts in atmospheric circulation[23,24] or in tropical climate modes[24,25]. The imprint of the D-O events is, nonetheless, reflected in large and globally synchronous changes in regional climates[26–28] transmitted through the atmospheric circulation everywhere except Antarctica and surrounding regions, where the signal is dominated by a slower oceanic response to changes in the north[29].

Ice-core records indicate that all of the D-O warmings were characterised by increased atmospheric $CO_2$, $CH_4$ and $N_2O$ concentrations[30–32], showing that these events had an impact on global biogeochemical cycles[4]. While it has been suggested that the reinvigoration of the AMOC during D-O warming events could itself result in the physical release of $CO_2$ to the atmosphere, diagnoses using a simple box model indicate the observed centennial-scale $CO_2$ change is largely a result of carbon release due to the warming[30]. D-O events provide an opportunity to quantify the warming-induced greenhouse-gas feedbacks to climate on a centennial timescale relevant to contemporary climate change. Here, we exploit this opportunity to provide new estimates for $CO_2$, $CH_4$ and $N_2O$ climate feedbacks.

## Feedback estimates from the Dansgaard-Oeschger events

The concept of feedback has been discussed in many previous studies, although terminologies differ[2,3] (see Methods for quantitative explanations). To estimate feedback strengths in terms of the associated change in radiative forcing (W m$^{-2}$) per degree (K) of global mean temperature change, we (a) identified the concentration changes in greenhouse gases from ice-core records across D-O events and converted them to radiative forcing; (b) used LOVECLIM model outputs to obtain the global mean temperature change during D-O events between 50 and 30 ka; and (c) combined both to derive feedback strengths, on the assumption that, on this timescale, the increase in global mean temperature leads to the increase in greenhouse gases.

Ice-core records of the concentration of $CO_2$ (ref. [30]), $CH_4$ (ref. [31]) and $N_2O$ (ref. [32]) during the period between 50 and 30 ka (Fig. 1, Supplementary Table 1) were converted to a common timescale (AICC2012) based on the age-depth relationships for each chronology[33]. We estimated the change in $CO_2$, $CH_4$ and $N_2O$ concentration associated with the warming phase of each D-O event (Supplementary Fig. 1.1–1.8), using the dating of the beginning of these events from ref. [14], which was also converted to the AICC2012 timescale. The concentration changes of the three greenhouse gases were converted to radiative forcing using equations given in ref. [34], as adopted by IPCC WG1 AR6 (ref. [35]), with concentration measurement uncertainties propagated into the corresponding radiative forcing uncertainties.

There are too few quantitative reconstructions of temperature changes, especially over land, to be able to make reliable estimates of changes in global mean temperature during the D-O events. We therefore use model-based estimates of the change in global mean temperature. The LOVECLIM model provides a global simulation of temperature changes during the interval 50–30 ka (ref. [36]) in response to realistic time-varying changes in orbital parameters, atmospheric trace gas concentrations and ice-sheet configuration, and by adding meltwater pulses at the correct times required to trigger each D-O event. Evaluation of the experiments against individual records[36,37] as well as comparison with the global compilation of palaeoclimate data in ref. [38] shows that it simulates the pattern of regional changes during individual D-O events during Marine Isotope Stage 3 well (Supplementary Fig. 2.1 to 2.8). We derived global mean temperature change by area-weighted averaging of the 64 × 32 grid cells, using the cosine of latitude as a weight (Fig. 2). The change in global mean temperature was identified in the same way as greenhouse gases (Supplementary Fig. 1.1–1.8).

The D-O events are not characterised by the ubiquitous warming of recent decades[39] since, although most of the land was warming, the ocean warmed in the northern hemisphere and cooled in the southern hemisphere (Supplementary Fig. 2.1 to 2.8). Nevertheless, overall both ocean and land temperatures increased on average (Supplementary Fig. 2.9) and the land/ocean warming ratio was 1.48 ± 0.08 (95 % CI), comparable to present-day warming[40,41]. The amplitude and rate of global mean temperature increase (Supplementary Table 2) were also comparable to those of present day, which is 0.95–1.20 K increase by the decade 2011 ~2020 compared to pre-industrial times (1850–1900) with a rate of 0.0068–0.0085 K/year[39]. These similarities mean that D-O events usefully constrain present-day greenhouse-gas climate feedbacks.

The value of feedback strength (in unit of W m$^{-2}$ K$^{-1}$) is the ratio of the radiative forcing brought about by the increases in $CO_2$, $CH_4$ and $N_2O$ to the increase in global mean temperature during D-O events (Fig. 3). A maximum likelihood method[42] is used to derive this ratio because it considers uncertainty of both the $x$- and $y$-variables (i.e. the driver and the response), in contrast with ordinary least squares regression which assigns uncertainty only to the $y$-variable. Based on the 8 D-O events that occurred between 50 and 30 ka, we estimated a feedback strength of 0.155 W m$^{-2}$ K$^{-1}$ for $CO_2$, 0.114 W m$^{-2}$ K$^{-1}$ for $CH_4$ and 0.106 W m$^{-2}$ K$^{-1}$ for $N_2O$, with standard errors of 0.018, 0.007 and 0.013, respectively (Table 1). The maximum likelihood method assumes that errors are normally distributed, so the 95% CIs can be obtained as ± 1.96 times the fitted standard error.

We also calculated the dimensionless quantity 'gain', a measure of the extent to which the change in global mean temperature would be reduced (if gain is positive) or increased (if gain is negative) in the absence of the feedback. Gains are estimated by multiplying the feedback strengths (W m$^{-2}$ K$^{-1}$) by the climate sensitivity parameter (K W$^{-1}$ m$^2$). Climate sensitivity (K) is

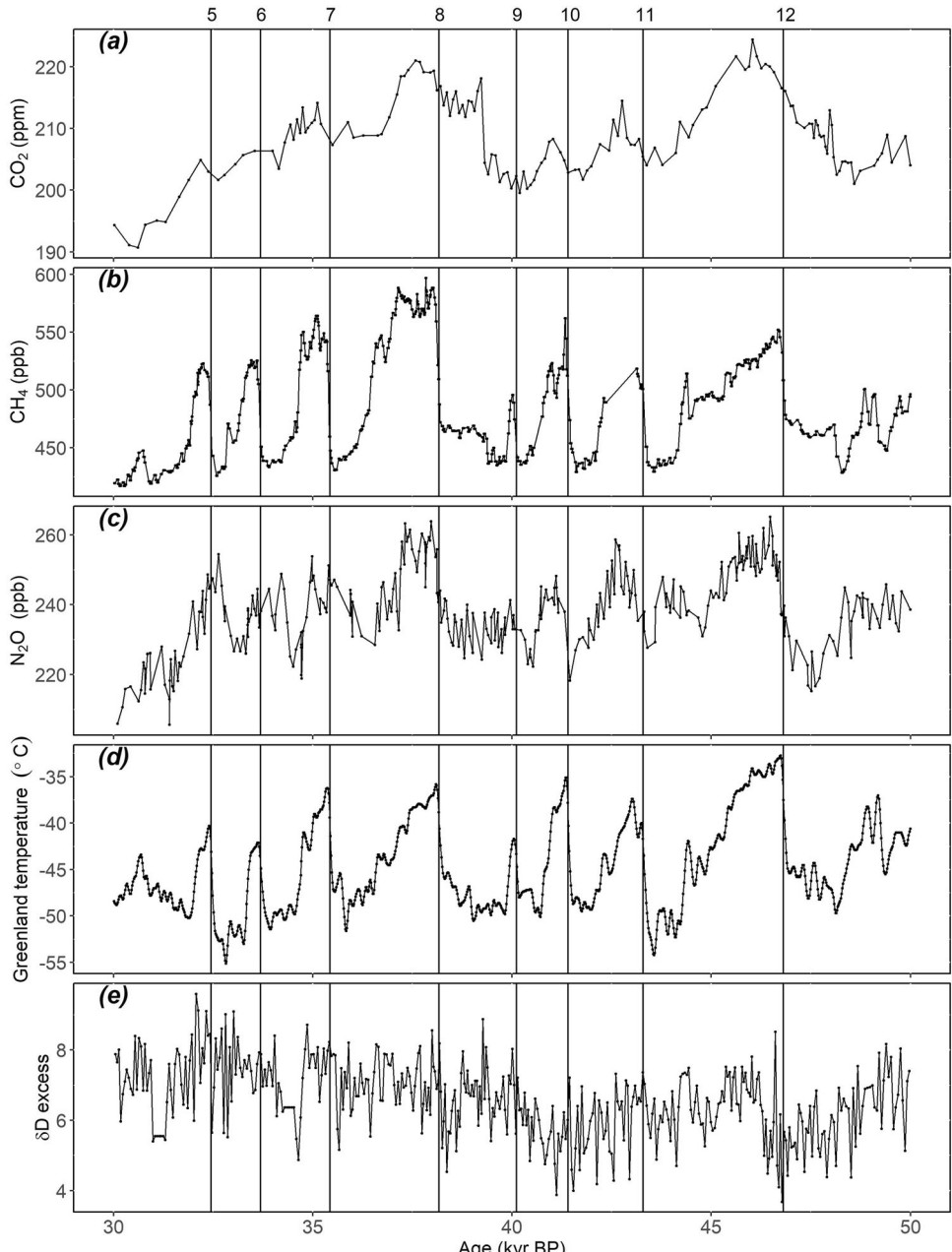

**Fig. 1 Ice-core records between 50 and 30 ka.** Changes in (**a**) $CO_2$, (**b**) $CH_4$, (**c**) $N_2O$, (**d**) Greenland temperature, and (**e**) δD excess. The vertical lines show the official start dates of the numbered D-O warming events. All data are on the AICC2012 timescale (BP 1950).

defined as the equilibrium temperature increase of the Earth's surface due to a radiative forcing equal to doubling atmospheric $CO_2$ concentration compared to the pre-industrial level, after all the fast physical climate feedbacks (but not ice sheets and greenhouse-gas concentrations) are taken into account. There are many recent estimates of this equilibrium climate sensitivity (ECS)[35,43–45]. We adopt the range derived from process-based assessment in IPCC WG1 AR6 (ref. [35]). Due to the non-normality of ECS, we directly use the corresponding net feedback parameter: in other words, the sum of all the fast physical climate feedbacks excluding those of ice sheets and greenhouse-gas concentrations (see Table 7.10 in ref. [35]). Assuming that the very likely range (−1.81 to −0.51 W m$^{-2}$ K$^{-1}$) can be treated as equivalent to a 90% confidence interval for the net feedback parameter, we derived a gain of 0.133 for $CO_2$, 0.099 for $CH_4$ and 0.091 for $N_2O$, with (approximate) standard errors of 0.048, 0.034

and 0.033, respectively (Table 1). We do not give confidence intervals for the gain as its distribution, as the ratio of two quantities assumed to be approximately normally distributed, is far from normal and in fact highly asymmetric (see Methods for details).

## Comparison with previous estimates

Model-based feedback estimates have been derived from simulations of the response to anthropogenic emissions, and separate the carbon-concentration feedback and the carbon-climate feedback[8]. Changes in the atmospheric carbon concentration caused by emissions are buffered by the land and ocean uptake through the carbon-concentration feedback (a negative feedback); the amount of carbon these sinks can absorb is reduced by the carbon-climate feedback (a positive feedback)[8]. In the present-

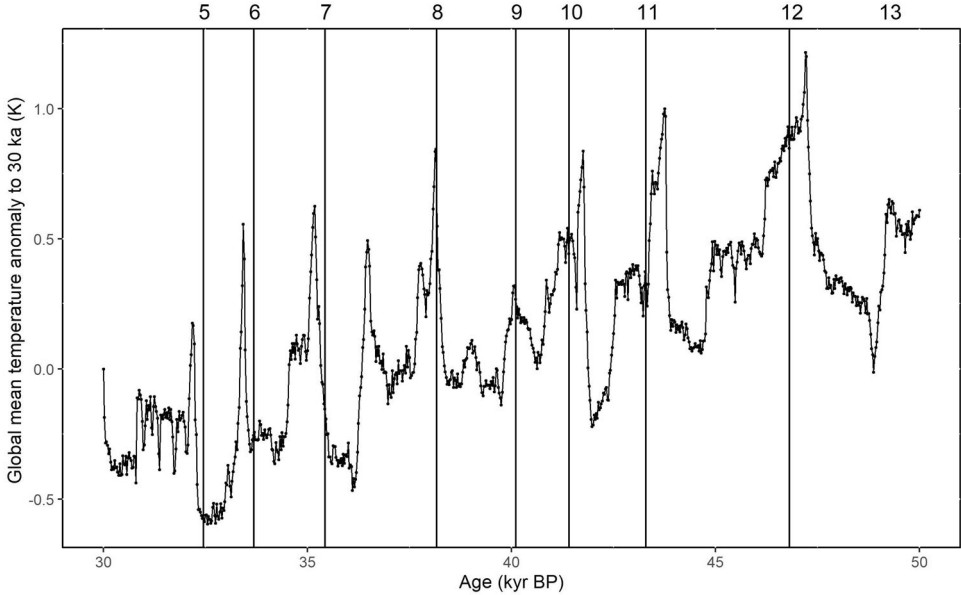

**Fig. 2 Global mean temperature anomaly to 30 ka.** The data were obtained from LOVECLIM simulations and binned in 25 years. The global mean temperature was area-weighted, using the cosine of latitude as a weight for each grid. The age is at absolute timescale. The vertical lines show the official start dates of the numbered D-O warming events on AICC2012 timescale (BP 1950).

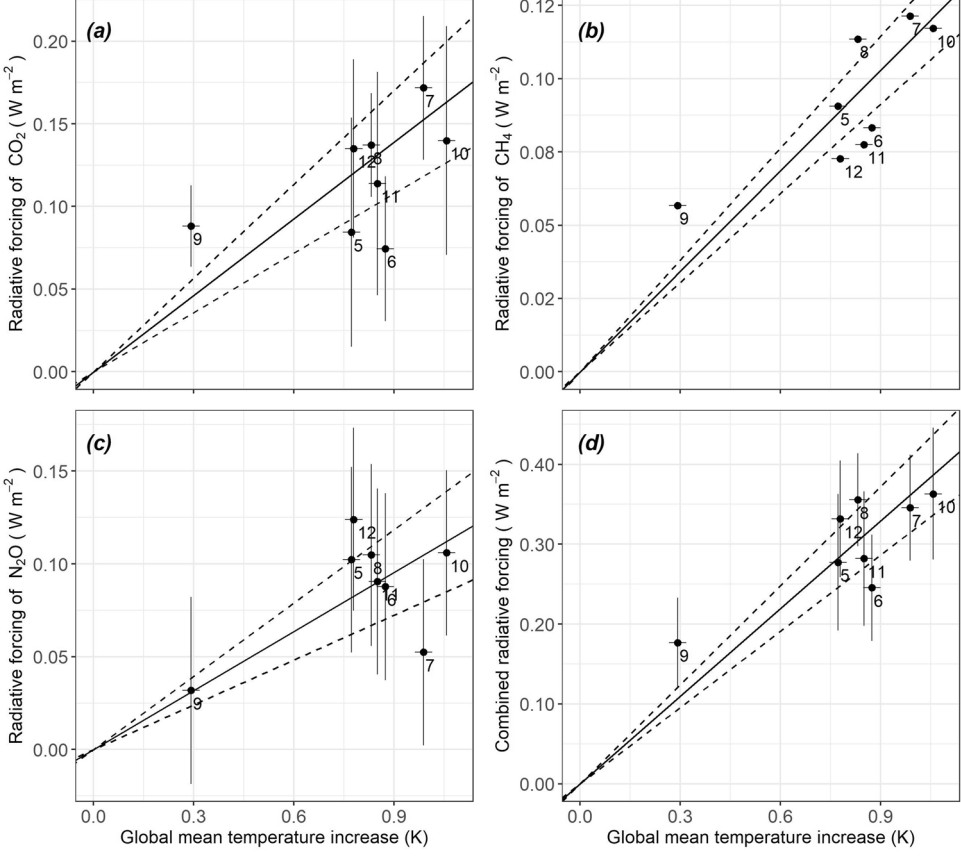

**Fig. 3 Maximum likelihood estimation of feedback strengths.** The figure shows the relationship between the increase in global mean temperature and radiative forcing induced by changes in (**a**) $CO_2$, (**b**) $CH_4$, (**c**) $N_2O$ concentrations and (**d**) combined radiative forcing of $CO_2$, $CH_4$ and $N_2O$. Each D-O event is numbered; the horizontal and vertical lines show the 95 % confidence intervals. The measurements of $CH_4$ concentration are very accurate so the vertical lines are too small to be observable on these plots. The solid line shows the maximum likelihood estimation of the ratio of radiative forcing to global mean temperature increase, the dashed lines show the 95 % confidence intervals of the ratio.

**Table 1 Feedbacks estimated from D-O events.**

|  | c (W m$^{-2}$ K$^{-1}$) | σ$_c$ (W m$^{-2}$ K$^{-1}$) | g | σ$_g$ |
|---|---|---|---|---|
| CO$_2$ | 0.155 | 0.018 | 0.133 | 0.048 |
| CH$_4$ | 0.114 | 0.007 | 0.099 | 0.034 |
| N$_2$O | 0.106 | 0.013 | 0.091 | 0.033 |
| Combined | 0.366 | 0.024 | 0.316 | 0.109 |

c is the feedback strength, g is the gain, while σ$_c$ is the standard error of the feedback strength and σ$_g$ is the standard error of the gain.

day context, anthropogenic CO$_2$ emissions are the main driver of changes in the carbon cycle and warming is the response of the emissions; in the D-O context, warming is the main driver and changes in the atmospheric CO$_2$ is the response. The feedback we quantified using D-O warmings equals the carbon-climate feedback defined in ref. [8]. See Supplementary Notes for a more detailed explanation.

Model estimates of the carbon-climate feedback based on simulations from the Coupled Climate Carbon Cycle Model Intercomparison (C$^4$MIP)[8] show considerable variability, with estimates ranging from 0.11 to 0.62 W m$^{-2}$ K$^{-1}$ (Fig. 4; see Supplementary Table 3). The range is somewhat reduced in models from the fifth and sixth phases of the Coupled Model Intercomparison Project (CMIP5[10], CMIP6[46]): 0.09 to 0.36 W m$^{-2}$ K$^{-1}$ in CMIP5 and −0.004 to 0.50 W m$^{-2}$ K$^{-1}$ in CMIP6. Our estimate of the CO$_2$ feedback derived from the D-O warming events is not consistent with high-end estimates from C$^4$MIP, CMIP5 and CMIP6, nor with low-end estimates from CMIP5 and CMIP6.

There are relatively few model-based estimates of the feedbacks associated with either CH$_4$ or N$_2$O (Fig. 4). IPCC WG1 AR6 (ref. [47]) estimated the CH$_4$-climate feedback due to the effect of temperature on methanogenesis in wetlands as 0.03 ± 0.01 W m$^{-2}$ K$^{-1}$ (1 standard deviation, based on limited evidence) and an additional, highly uncertain feedback of 0.01 (0.003 to 0.04, 5th to 95th percentile range, also based on limited evidence) W m$^{-2}$ K$^{-1}$ due to permafrost thaw. Our results suggest that the CH$_4$-climate feedback is larger than that assessed by AR6. Xu-Ri et al.[9] simulated terrestrial N$_2$O feedback estimate to be 0.11 W m$^{-2}$ K$^{-1}$. This is within the range estimated from the D-O warming events. Stocker et al.[11] estimated the terrestrial feedbacks associated with CO$_2$, CH$_4$ and N$_2$O to be 0.079, 0.011 and 0.023 W m$^{-2}$ K$^{-1}$ using the LPX-Bern vegetation model. IPCC WG1 AR6 (ref. [47]) estimated the land N$_2$O feedback as 0.02 ± 0.01 W m$^{-2}$ K$^{-1}$ (with low confidence) and the oceanic N$_2$O feedback as −0.008 ± 0.002 W m$^{-2}$ K$^{-1}$ (based on limited evidence). Thus, AR6 indicates that the total N$_2$O feedback is positive and dominated by the land, while the ocean feedback is smaller and of opposite sign. The combined (land plus ocean) feedback strength for N$_2$O according to AR6 ((0.02 − 0.008) ± √(0.01$^2$ + 0.002$^2$) = 0.012 ± 0.010 W m$^{-2}$ K$^{-1}$) however is considerably smaller than the value indicated by the D-O records.

Modern observations have been used to constrain model-based estimates of biosphere feedbacks. Gedney et al.[7] used multi-site flux measurements as a constraint on simulated wetland CH$_4$ emissions to obtain feedback estimates in the range of 0.01 to 0.11 W m$^{-2}$ K$^{-1}$ (Fig. 4). Other studies have used the emergent constraint approach to estimate the sensitivity of tropical land carbon storage to warming[48,49], but only address part of the CO$_2$ feedback and cannot be used to derive a comparison. This lack of strong observational constraints has motivated the use of past climate changes to estimate greenhouse-gas feedbacks to climate[50–52].

Previous attempts to quantify greenhouse-gas feedbacks using past climate changes have focused on the volcanically forced cooling during the Little Ice Age (LIA: 1500-1750 CE) which was associated with a decrease in CO$_2$ of ca 8 ppm[53]. However, these estimates vary considerably and have high uncertainties (Fig. 4), in part associated with the temperature reconstruction used and in part due to differences in methodology. Scheffer et al.[51] used alternative reconstructions of the LIA temperature change, derived from Mann and Jones[54] and Moberg et al.[55], and obtained estimates of 1/(1 − g$_{CO2}$) of 1.28–2.93 and 1.07–1.25 (corresponding to a feedback of 0.54 ± 0.27 W m$^{-2}$ K$^{-1}$ and 0.16 ± 0.08 W m$^{-2}$ K$^{-1}$, respectively). Cox and Jones[52] obtained an estimate of 40 ± 20 ppm CO$_2$ per K, using the Moberg et al. reconstruction[55], which corresponds to a feedback of 0.53 ± 0.26 W m$^{-2}$ K$^{-1}$. Our recalculation of the CO$_2$ feedback using the full 7000-member ensemble of temperature reconstructions provided by the PAGES2k Consortium[56] produced a lower estimate than either Scheffer et al.[51] using the Mann and Jones reconstruction[54] or Cox and Jones[52], but still with very large uncertainties that encompass almost all of the previous LIA estimates (Fig. 4). This uncertainty is also seen in recalculations of the feedback associated with changes in CH$_4$ and N$_2$O over the LIA, suggesting that the LIA does not provide a sufficiently strong constraint to provide reliable estimates. In contrast, the D-O warming events provide a strong constraint because the temperature changes, and the responses, are relatively large. Furthermore, replication over 8 events considerably reduces uncertainty compared to using a single event such as the LIA.

We rely on the LOVECLIM model to derive estimates of global temperature because there are insufficient observationally based, quantitative reconstructions to estimate these reliably. Although a number of modelling groups have made simulations to mimic D-O events during the glacial by adding freshwater forcing[57–61], none of these have used realistic forcings for individual D-O events. Comparison of the spatial patterns of the LOVECLIM simulated temperature changes for individual D-O events with records from the Voelker data compilation[38] (Supplementary Fig. 2.1–2.8) indicate that there is good qualitative agreement in the sign of the change, with >75% of the grid cells being correctly predicted (Supplementary Table 4). Although LOVECLIM is a low-resolution model and the simulations were made with fixed cloud cover, neither of these constraints should have a major impact on the estimates of global temperature[62]. Furthermore, analyses based on estimating global temperature from observed temperature changes in Greenland over the interval between 80 and 20 ka using the relationship between simulated Greenland and global temperature obtained from the LOVECLIM simulations (see Supplementary Notes) produce comparable estimates of feedback strength. Thus, although the use of model outputs is a potential source of additional uncertainty, in the absence of a compelling alternative this approach provides a way to estimate greenhouse-gas climate feedbacks on centennial scales.

We have assumed that there is a strong relationship between global temperature changes and greenhouse-gas emissions during D-O warming events in order to estimate the climate feedback. Some of the changes in emissions may reflect change in hydroclimate, particularly in tropical regions[27], but we presume that such changes are also conditioned by changes in temperature and thus will be reflected in the global temperature record. Similarly, changes in the balance of marine versus terrestrial sources of greenhouse-gas emissions, particularly CO$_2$, are influenced by the changes in global temperature. There is currently insufficient information to disentangle the regional sources of greenhouse-gas

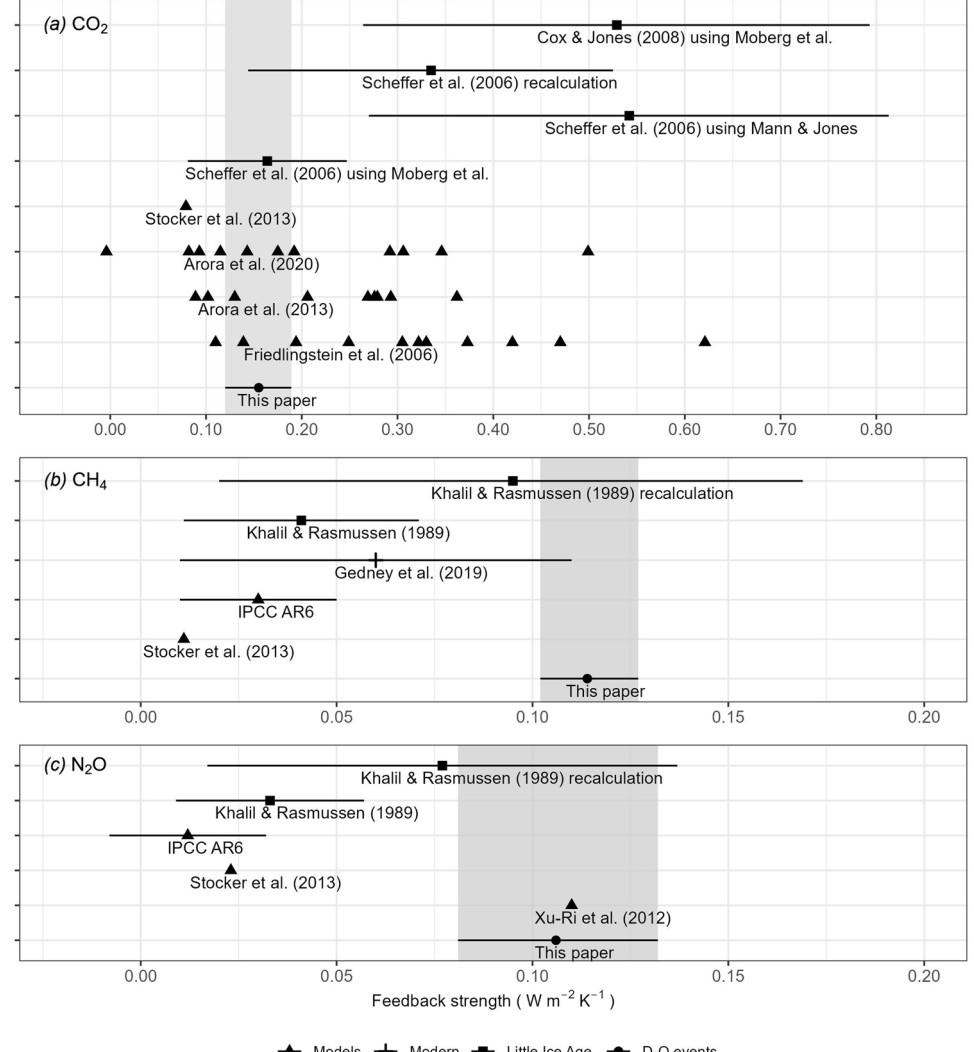

**Fig. 4 Comparison of feedback strengths.** Feedbacks of this paper and previous estimates for (**a**) $CO_2$, (**b**) $CH_4$ and (**c**) $N_2O$. The shaded bars show the results from this paper (95% CI). Horizontal lines show the range on each previous estimate. All the models estimate feedbacks at 2100. Stocker et al.[11] only simulates the land climate feedbacks. The recalculated estimates for the Little Ice Age are based on the full 7000-member ensemble of global mean temperature reconstructions provided by the PAGES2k Consortium, using the 95% range to approximate 95% confidence intervals.

emissions during the D-O events. However, the global feedback estimates obtained from analysis of the D-O events indicates that these feedbacks are non-negligible and poorly represented in current models.

## Conclusions

We have used D-O cycles to estimate the climate feedbacks associated with $CO_2$, $CH_4$ and $N_2O$. These feedbacks would amplify the equilibrium global mean temperature increase by about 15%, 11% and 10%, respectively (calculated as 1/(1–gain) – 1). The combined feedback from changes in all three greenhouse gases is about 46%.

## Methods

**Quantitative explanation of feedback terms.** The concept of feedback has been explained quantitatively in many previous studies, although terminologies differ[2,3,63,64]. Briefly, a perturbation to the energy balance of a system, termed radiative forcing, pushes the system to a new equilibrium state with a change in temperature[2,3,63]. A reference system without feedbacks, gives a temperature increase ($\Delta T_0$) with a radiative forcing ($\Delta R_0$) when it reaches equilibrium; the ratio of $\Delta T_0$ to $\Delta R_0$, denoted $\lambda_0$, is the climate sensitivity parameter of this

reference system[2,3,63].

$$\Delta T_0 = \lambda_0 \Delta R_0 \quad (1.1)$$

Feedbacks results in additional radiative forcing. The temperature increase at equilibrium with feedback ($\Delta T$) is thus:

$$\Delta T = \lambda_0 \left( \Delta R_0 + \Delta R_1 + \Delta R_2 + \dots + \Delta R_n \right) \quad (1.2)$$

Assuming $\Delta R_1, \Delta R_2, \dots, \Delta R_n$ proportional to $\Delta T$ with parameters $c_1, c_2, \dots, c_n$ (refs. [2,3]) gives:

$$\Delta T = \lambda_0 \left( \Delta R_0 + c_1 \Delta T + c_2 \Delta T + \dots + c_n \Delta T \right) \quad (1.3)$$

Combining Eqs. 1.1, 1.3 gives:

$$\Delta T = \Delta T_0 + \lambda_0 c_1 \Delta T + \lambda_0 c_2 \Delta T + \dots + \lambda_0 c_n \Delta T \quad (1.4)$$

The terms $c_1, c_2, \dots, c_n$ express feedbacks in radiative forcing per degree of temperature increase ($W\,m^{-2}\,K^{-1}$). This metric can be converted to a dimensionless measure called gain ($g_1, g_2, \dots, g_n$), by multiplying by $\lambda_0$ (refs. [2,3]):

$$\Delta T = \Delta T_0 + g_1 \Delta T + g_2 \Delta T + \dots + g_n \Delta T \quad (1.5)$$

The relationship between the equilibrium temperature increase with and without feedback is thus:

$$\Delta T = \frac{\Delta T_0}{(1 - g_1 - g_2 - \dots - g_n)} = \frac{\Delta T_0}{(1 - g_{total})} \quad (1.6)$$

Equation 1.6 shows that: (a) a positive gain amplifies $\Delta T_0$ and a negative gain diminishes $\Delta T_0$; (b) the gain shows by what fraction $\Delta T_0$ is less than $\Delta T$; a gain of 0.2, for example, means that $\Delta T_0$ is 20% less than $\Delta T$, which means that $\Delta T$ is 25% more than $\Delta T_0$; (c) independent gains sum to $g_{total}$, but their impacts on amplifying

$\Delta T_0$ are not additive; two gains of 0.2, for example, combine to make $\Delta T$ 67% more than $\Delta T_0$ (refs. [2,3]).

**Ice-core data.** We used the ice-core records of atmospheric $CO_2$, $CH_4$ and $N_2O$ concentrations detailed in Supplementary Table 1. The age models were converted to the Antarctic Ice-Core Chronology 2012 (AICC2012) timescale[33] prior to analysis. The average resolution of the records on the AICC2012 timescale is 134 years for $CO_2$, 18 years for $CH_4$, and 59 years for $N_2O$ over the period 50–30 ka. Greenland temperatures were taken from the NGRIP ice core[65], and again the original chronology was converted to the AICC2012 chronology before analysis. The average resolution for Greenland temperature is 19 years. We compare this with the δD excess record from the EPICA Dome C (EDC) ice core[66], which has an average resolution of 49 years. Strictly speaking δD excess is interpreted as temperature changes in the source area rather than global temperature[67]. Nevertheless, it does clearly show the temperature changes associated with the D-O events.

The conversion to the AICC2012 chronology introduces additional uncertainties to those inherent in the original ice-core age models, particularly for the earlier part of the record[33]. Nevertheless, these are unlikely to have a remarkable effect given the method of determining the minimum and maximum response and thus of estimating the amplitude of change.

**LOVECLIM temperature simulations.** We used a transient simulation of the interval 50–30 ka performed with the LOVECLIM model[36] to obtain an estimate of global mean temperature during the D-O events. LOVECLIM is a computationally efficient low-resolution (horizontal resolution of the atmospheric model is 5.625°) global climate model. The model was spun-up to equilibrium using an initial atmospheric $CO_2$ concentration of 207.5 ppm, orbital forcing appropriate for 50 ka BP and an estimate of the 50 ka ice-sheet orography and albedo obtained from an off-line ice-sheet model simulation[68]. The transient run was initialised from this spin-up and run from 50 to 30 ka. Orbital, greenhouse gas, and ice-sheet forcings were updated continuously during the transient simulation; orbital parameters were derived following ref. [69], greenhouse-gas concentrations were from ice-core records, and the ice-sheet was from the off-line ice-sheet model simulation. In order to trigger D-O events, a time-series of freshwater inputs was derived by optimising freshwater fluxes such that simulated sea-surface temperature (SST) in the eastern subtropical North Atlantic were congruent with alkenone-based reconstructions of SST in that region.

We used simulated atmospheric temperature from the LOVECLIM experiments. Analyses in the original paper[36] indicate that the simulations reproduce broadscale features of climate change during the D-O cycles well, and there is a good match with quantitative estimates for specific D-O events (e.g. D-O 8) from the Iberian margin and western Mediterranean regions, where highly resolved SST records are available[36,37]. The simulated air temperature changes over Greenland are somewhat smaller than those inferred from Greenland ice-core records[36].

The LOVECLIM simulations were run with fixed cloud cover in these hindcast experiments. Studies examining the impact of using fixed clouds, albeit in a different model[62], suggest that changes in cloud cover accentuate the temperature changes: it gets colder in the Northern Hemisphere, particularly in the North Atlantic region, but warmer in the Southern Hemisphere. However, the enhanced Northern Hemisphere cooling and Southern Hemisphere warming compensate each other so that the impact on global mean temperature is small. We assume that the same would be true in the LOVECLIM simulations.

To further evaluate the reliability of the LOVECLIM simulations, we compared the simulated temperature changes to reconstructions from the Voelker data set[38], the only global data set that currently exists for MIS 3. Since this data set only contains a few records with quantitative estimates at high enough resolution to identify the temperature change for each D-O event, we compared the geographic patterns in warming or cooling trends globally (Supplementary Fig. 2.1–2.8; Supplementary Table 4). This analysis shows that: (a) the D-O events are registered as warmings over nearly all of the land areas in the world; (b) the geographic patterns of warming or cooling trends are consistent between observations and simulations, accepting that there may not be an exact geographic mapping because of the low resolution of the model; and (c) where there is quantitative information, the results are broadly consistent with the magnitude of the simulated changes.

Menviel et al.[37] also showed that LOVECLIM surface air temperature and sea-surface temperature anomalies for D-O stadials and Heinrich stadials are consistent with observations, which provides further confidence that the LOVECLIM model captures global temperature change patterns linked to these events.

**Identification of minimum and maximum.** We used the start date of each D-O event provided by ref. [14]. We then calculated binned averages of the $CO_2$, $CH_4$, $N_2O$ records and LOVECLIM simulated global mean temperature anomaly to 30 ka, centred on each D-O start date, using 25-year bins.

There is some uncertainty in the chronology of the start dates of each D-O event, and further uncertainty may be caused by the conversion from the GICC05 to the AICC2012 timescale (Supplementary Table 1). We therefore used a 200-year interval before and after the assumed D-O start date to identify the minima for $CO_2$, $CH_4$, $N_2O$ and LOVECLIM simulated global mean temperature anomaly to 30 ka. We assumed the maxima must occur within 500 years for $CO_2$, $CH_4$ and

LOVECLIM simulated global mean temperature anomaly to 30 ka, and 600 years for $N_2O$. The different lengths of time considered reflect the time needed to reach equilibrium and are also influenced by the resolution of the records. See Supplementary Fig. 1.1–1.8 for details for each D-O event.

**Calculation of radiative forcing and propagation of uncertainties.** We calculated binned values of each gas as follows:

$$c_{gas} = \frac{\sum_{k=1}^{m} c_{gas,k}}{m} \tag{5.1}$$

$$\sigma_{c_{gas}} = \sqrt{\frac{\sum_{k=1}^{m} \sigma_{c_{gas,k}}^2}{m^2}} \tag{5.2}$$

where $c_{gas,k}$ denotes a value in the bin with its standard error $\sigma_{cgas,k}$; $m$ denotes the total number of values in this bin; $c_{gas}$ denotes the average value in this bin with its propagated standard error $\sigma_{cgas}$.

We calculated the radiative forcing[34] associated with the change between minimum and maximum values for each event, as follows:

$CO_2$:

$$\Delta R_C = \left(a_1(C - C_0)^2 + b_1|C - C_0| + c_1\bar{N} + 5.36\right) \times \ln\left(\frac{C}{C_0}\right) \tag{5.3}$$

$$\sigma_{\Delta R_C} = \sqrt{\left(\frac{\partial \Delta R_C}{\partial C}\right)^2 \sigma_C^2 + \left(\frac{\partial \Delta R_C}{\partial C_0}\right)^2 \sigma_{C_0}^2} \tag{5.4}$$

where $a_1 = -2.4 \times 10^{-7}$ W m$^{-2}$ ppm$^{-1}$, $b_1 = 7.2 \times 10^{-4}$ W m$^{-2}$ ppm$^{-1}$, $c_1 = -2.1 \times 10^{-4}$ W m$^{-2}$ ppb$^{-1}$

$CH_4$:

$$\Delta R_M = \left(a_2\bar{M} + b_2\bar{N} + 0.043\right) \times \left(\sqrt{M} - \sqrt{M_0}\right) \tag{5.5}$$

$$\sigma_{\Delta R_M} = \sqrt{\left(\frac{\partial \Delta R_M}{\partial M}\right)^2 \sigma_M^2 + \left(\frac{\partial \Delta R_M}{\partial M_0}\right)^2 \sigma_{M_0}^2} \tag{5.6}$$

where $a_2 = -1.3 \times 10^{-6}$ W m$^{-2}$ ppb$^{-1}$, $b_2 = -8.2 \times 10^{-6}$ W m$^{-2}$ ppb$^{-1}$

$N_2O$:

$$\Delta R_N = \left(a_3\bar{C} + b_3\bar{N} + c_3\bar{M} + 0.117\right) \times \left(\sqrt{N} - \sqrt{N_0}\right) \tag{5.7}$$

$$\sigma_{\Delta R_N} = \sqrt{\left(\frac{\partial \Delta R_N}{\partial N}\right)^2 \sigma_N^2 + \left(\frac{\partial \Delta R_N}{\partial N_0}\right)^2 \sigma_{N_0}^2} \tag{5.8}$$

where $a_3 = -8.0 \times 10^{-6}$ W m$^{-2}$ ppm$^{-1}$, $b_3 = 4.2 \times 10^{-6}$ W m$^{-2}$ ppb$^{-1}$, $c_3 = -4.9 \times 10^{-6}$ W m$^{-2}$ ppb$^{-1}$ $C, M, N$ denote the maximum values identified for $CO_2$, $CH_4$ and $N_2O$, respectively; $C_0, M_0, N_0$ denote the minimum values identified for $CO_2$, $CH_4$ and $N_2O$, respectively; $\bar{C}, \bar{M}, \bar{N}$ denote the mean values identified for $CO_2$, $CH_4$ and $N_2O$, respectively; $\Delta R_C, \Delta R_M, \Delta R_N$ denote the radiative forcing brought about by $CO_2$, $CH_4$ and $N_2O$, with their corresponding standard errors, $\sigma_{\Delta RC}$, $\sigma_{\Delta RM}$, $\sigma_{\Delta RN}$, respectively.

**Calculation of temperature increase and propagation of uncertainties.** LOVECLIM provides yearly outputs, we used the standard deviation in each 25-year bin to approximate the standard error of the binned average.

The global mean temperature and its standard error was calculated as follows:

$$T_{mean\ global} = \frac{\sum T_{each} w_{each}}{\sum w_{each}} \tag{6.1}$$

$$\sigma_{T_{mean\ global}} = \frac{\sqrt{\sum \sigma_{T_{each}}^2 w_{each}^2}}{\sum w_{each}} \tag{6.2}$$

where the weight of each grid ($w_{each}$) is the cosine value of the latitude (in radian) of that grid.

We converted the data to anomaly to 30 ka as in the original paper[36]:

$$T_{mean\ global\ anomaly} = T_{mean\ global} - T_{mean\ global,30\ ka} \tag{6.3}$$

$$\sigma_{T_{mean\ global\ anomaly}} = \sqrt{\sigma_{T_{mean\ global}}^2 + \sigma_{T_{mean\ global,30\ ka}}^2} \tag{6.4}$$

The global mean temperature change and its standard error were calculated using the minimum and maximum identified for each D-O event:

$$\Delta T_{mean\ global} = T_{mean\ global\ anomaly,max} - T_{mean\ global\ anomaly,min} \tag{6.5}$$

$$\sigma_{\Delta T_{mean\ global}} = \sqrt{\sigma_{T_{mean\ global\ anomaly,max}}^2 + \sigma_{T_{mean\ global\ anomaly,min}}^2} \tag{6.6}$$

**Calculation of gain and propagation of uncertainties.**

$$g = c\lambda_0 \tag{7.1}$$

$$\sigma_g = \sqrt{c^2 \sigma_{\lambda_0}^2 + \lambda_0^2 \sigma_c^2} \tag{7.2}$$

where $g$ is the gain; $\sigma_g$ is the standard error of the gain; $c$ is the maximum likelihood estimated slope from the Deming package[70], using radiative forcing ($\Delta R_C$, $\Delta R_M$, $\Delta R_N$, $\Delta R_C + \Delta R_M + \Delta R_N$) and temperature increase ($\Delta T_{mean\ global}$) with the standard error of radiative forcing ($\sigma_{\Delta RC}$, $\sigma_{\Delta RM}$, $\sigma_{\Delta RN}$, $\sqrt{(\sigma^2_{\Delta RC} + \sigma^2_{\Delta RM} + \sigma^2_{\Delta RN})}$) and the standard error of temperature increase ($\sigma_{\Delta Tmean\ global}$) as inputs, with the intercept set to 0; $\sigma_c$ is the standard error of $c$ obtained using the Deming package[70]; $\lambda_0$ is the climate sensitivity parameter; $\sigma_{\lambda 0}$ is the standard error of $\lambda_0$.

We note that the climate sensitivity parameter ($\lambda_0$) is calculated from $\Delta F_{2 \times CO2}$ and ECS:

$$\lambda_0 = \frac{ECS}{\Delta F_{2 \times CO2}} \tag{7.3}$$

while ECS is calculated from $\Delta F_{2 \times CO2}$ and the net feedback parameter ($\alpha_{net}$), according to chapter 7 in IPCC WG1 AR6 (ref. [35]):

$$ECS = -\frac{\Delta F_{2 \times CO2}}{\alpha_{net}} \tag{7.4}$$

Combining Eqs. (7.3) and (7.4) gives

$$\lambda_0 = -\frac{1}{\alpha_{net}} \tag{7.5}$$

The error propagation rule requires the input variables to be at least approximately normally distributed. The distribution of ECS is not normal because it is a ratio of two normally distributed quantities ($\Delta F_{2 \times CO2}$, $\alpha_{net}$) with a non-zero centre. However, $\alpha_{net}$ can reasonably be assumed to be normal (see the first paragraph of chapter 7.4.2.7 in IPCC WG1 AR6 (ref. [35])). Therefore, it can be used to calculate $\lambda_0$. Its standard error is

$$\sigma_{\lambda_0} = \left| \frac{\lambda_0\ \sigma_{\alpha_{net}}}{\alpha_{net}} \right| \tag{7.6}$$

We adopt the very likely range given in Table 7.10 in ref. [35] as the 90% confidence interval, then following the above equations we obtain $\lambda_0 = 0.86\ \text{K W}^{-1}\ \text{m}^2$ and $\sigma_{\lambda 0} = 0.29\ \text{K W}^{-1}\ \text{m}^2$.

**Calculation of gains from previous estimates.** Some of the previous estimates give gains directly[8,10]; some give the amplifications[51], $1/(1 - \text{gain})$, which can be converted to gains easily; some provide values of $c$ (radiative forcing per degree)[7,9,11], which can be converted to gains using Eqs. 7.1, 7.2; some give the $CO_2$ concentration gradient ($\partial CO_2 / \partial T$; unit: ppm/K)[52], which can be approximated to gains using Eqs. 8.1, 8.2.

$$g \approx \frac{\partial CO_2}{\partial T}\ d_0 \tag{8.1}$$

$$\sigma_g \approx \sqrt{\left(\frac{\partial CO_2}{\partial T}\right)^2 \sigma^2_{d_0} + d_0^2\ \sigma^2_{\frac{\partial CO_2}{\partial T}}} \tag{8.2}$$

where $d_0$ can be obtained from $\lambda_0$ by multiplying a conversion factor (3.7 W m$^{-2}$/ 280ppm).

Some previous estimates give the minimum and maximum concentration of gases and northern hemisphere temperature change during Little Ice Age (LIA)[50], which can be converted to corresponding radiative forcing and global mean temperature change, assuming the global mean temperature change to be 2/3 of the northern hemisphere temperature change. There is only one estimate for the LIA, so maximum likelihood estimation of the slope is not available. Instead, we use Eqs. 8.3, 8.4 to derive $c$ values, then use Eqs. 7.1, 7.2 to derive gains.

$$c = \Delta R / \Delta T \tag{8.3}$$

$$\sigma_c = c \sqrt{\left(\frac{\sigma_{\Delta R}}{\Delta R}\right)^2 + \left(\frac{\sigma_{\Delta T}}{\Delta T}\right)^2} \tag{8.4}$$

where $c$ is radiative forcing per kelvin; $\sigma_c$ is the standard error of $c$; $\Delta R$ is the radiative forcing; $\sigma_{\Delta R}$ is the standard error of $\Delta R$; $\Delta T$ is global mean temperature change; $\sigma_{\Delta T}$ is the standard error of $\Delta T$.

Previous LIA estimates use either the Moberg et al.[55] or Mann and Jones[54] climate reconstructions; neither can now be assumed to be accurate. We therefore recalculated the feedbacks using the full 7000-member ensemble across all methods of the PAGES2k Consortium 2019 global mean temperature reconstructions[56]. Assuming the 95% range as an approximation of the 95% confidence interval, we derive a global mean temperature change ($\Delta T$) with a standard error ($\sigma_{\Delta T}$). We identified the minimum and maximum concentration of the greenhouse gases with their standard errors from the same data source as in the LIA feedback papers, and converted these to radiative forcing using the same method as used for D-O events. Finally, we used Eqs. 8.3, 8.4 to obtain $c$ values, then used Eqs. 7.1, 7.2 to obtain gains.

## Data availability
LOVECLIM model outputs for temperature can be downloaded from http://apdrc.soest. hawaii.edu/las/v6/dataset?catitem=0 by choosing Datasets > APDRC Public-Access

Products > Paleoclimate modelling > LOVECLIM > Dansgaard-Oeschger > surface temperature. Other datasets used and generated during this study, are compiled in the public GitHub repository, https://github.com/ml4418/Greenhouse-gas-climate-feedback-paper.git.

## Code availability
The R scripts used in this study are available in the public GitHub repository, https://github.com/ml4418/Greenhouse-gas-climate-feedback-paper.git.

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

## Acknowledgements

S.P.H. acknowledges support from the European Research Council (ERC) for GC2.0: Unlocking the past for a clearer future (ERC-2015-AdG 694481) and from the JPI-Belmont project PAlaeo-Constraints on Monsoon Evolution and Dynamics (PAC-MEDY) through the UK Natural Environmental Research Council (NERC: NE/P006752/1). I.C.P. acknowledges funding from the ERC, under the European Union's Horizon 2020 research and innovation programme (grant agreement No: 787203 REALM). M.L. acknowledges support from Imperial College through the Lee Family Scholarship. L.M. acknowledges support from the Australian Research Council (grant FT180100606). We thank Eric Wolff for advice about the ice-core chronologies. This work is a contribution to the Imperial College initiative on Grand Challenges in Ecosystems and the Environment.

## Author contributions

M.L., I.C.P. and S.P.H. were responsible for the analyses. L.M. provided the LOVECLIM model outputs. M.L. and S.P.H. wrote the first draft of the paper, and all authors contributed to the final draft.

## Competing interests

The authors declare no competing interests.
