## [Peer Review File · Communications Earth & Environment]

Web links to the author's journal account have been redacted from the decision letters as indicated to maintain confidentiality.

6th Oct 21

Dear Ms Liu,

Firstly, I would like to sincerely apologise once again for the extremely long delay in sending a decision on your manuscript titled "Past rapid warmings as a constraint on greenhouse-gas climate feedbacks". This was due to difficulty in assigning reviewers, compounded by the need to replace a reviewer after an extended delay in receiving their report. Your manuscript has now been seen by 3 reviewers, whose comments are appended below. You will see that they find your work of some potential interest. However, they have raised quite substantial concerns that must be addressed. In light of these comments, we cannot accept the manuscript for publication, but would be interested in considering a revised version that fully addresses these serious concerns.

In particular, please ensure that the revised manuscript meets the following editorial thresholds:

****Provide a compelling case that your modelling approach is robust, including a thorough justification of the fundamental assumptions or, alternatively refocus the manuscript to ensure that your conclusions are fully supported by your modelling****

****Ensure that your methods are sufficiently described so as to enable the assessment and reproducibility of your study****

We hope you will find the reviewers' comments useful as you decide how to proceed. Should additional work allow you to address these criticisms, we would be happy to look at a substantially revised manuscript.

However, please bear in mind that we will be reluctant to approach the reviewers again in the absence of substantial revisions.

If the revision process takes significantly longer than three months, we will be happy to reconsider your paper at a later date, as long as nothing similar has been accepted for publication at Communications Earth & Environment or published elsewhere in the meantime.

We understand that due to the current global situation, the time required for revision may be longer than usual. We would appreciate it if you could keep us informed about an estimated timescale for resubmission, to facilitate our planning. Of course, if you are unable to estimate, we are happy to accommodate necessary extensions nevertheless.

Please use the following link to submit your revised manuscript, point-by-point response to the referees' comments (which should be in a separate document to any cover letter) and any completed checklist:

[link redacted]

Please do not hesitate to contact me if you have any questions or would like to discuss the required revisions further. Thank you for the opportunity to review your work.

Best regards,

Joe Aslin

Associate Editor,
Communications Earth & Environment
<https://www.nature.com/commsenv/>
Twitter: @CommsEarth

EDITORIAL POLICIES AND FORMAT

If you decide to resubmit your paper, please ensure that your manuscript complies with our editorial policies and complete and upload the checklist below as a Related Manuscript file type with the revised article:

Editorial Policy Policy requirements

For your information, you can find some guidance regarding format requirements summarized on the following checklist:(<https://www.nature.com/documents/commsj-phys-style-formatting-checklist-article.pdf>) and formatting guide (<https://www.nature.com/documents/commsj-phys-style-formatting-guide-accept.pdf>).

REVIEWER COMMENTS:

Reviewer #1 (Remarks to the Author):

The manuscript submitted by Liu et al. to Communications Earth & Environment aims to quantify the feedback strength of increasing greenhouse gases (GHG) in the warming amplitude of Dansgaard-Oeschger (D-O) events. Studying the individual feedbacks of increasing CO₂, CH₄ and N₂O on climate during these past global and analogues events, in time and magnitude, to the present warming is a clever approach to tackle the strength of the increasing GHG concentration in the current warming. This approach circumvents the lack of modern records spanning decadal to centennial time periods and the feedback uncertainties due to the anthropogenic impact. Furthermore, Earth System Models used to estimate the feedback strength are hampered by the lack of many biosphere processes. The feedback estimates from D-O events were based on a) converting GHG concentration changes from ice archives to radiative forcing using the equations given by the IPCC (2013), b) inferring global mean temperatures from Greenland temperature records using the LOVECLIM model, and c)

combining both to derive feedback strength. This methodology is relevant and the conclusions presented in the manuscript are worth to be published in Comm. Earth & Environment. They challenge previous estimates on the strength of GHG feedback and, particularly, for CO₂ values. I have only few comments/suggestions that the authors should address before the publication of this work.

- Lines 77-78: to reinforce their assumption that increase in global temperature leads the increase in GHG concentration it would be interesting to add in Figure 1 the curve of δD , a proxy for Antarctic temperature changes that are interpreted as global temperature variations.

- Lines 79-80 and caption of Figure 1: why does the CO₂ concentration record only go back to 67 ka ?

- Lines 91-96 / lines 233-263: The database of ref 37 is not updated and is mainly a compilation of sea surface temperature records with very few terrestrial ones. If the pattern of regional D-O warming events is relatively well reproduced by the LOVECLIM model, the magnitude of the warming is largely underestimated for the eastern part of the North Atlantic ocean and the western Mediterranean Sea. For instance, in the Alboran Sea, Gulf of Lyon and southwestern Iberian margin the SST increase during the D-O 8 and 12 is of $\sim 9^{\circ}\text{C}$ (Cacho et al., *Paleoceanography*, 1999 ; Paillet and Bard, *Paleo* 3, 2002 ; Català et al., *Climate of the Past*, 2019) while the simulated SST only show an increase of 2-4 $^{\circ}\text{C}$ (Fig. S2.4, Fig. S2.8). In the northwestern Iberian margin and the Bay of Biscay, the estimated SST increase for D-O warming is even larger while in the model the increase remains at 2-4 $^{\circ}\text{C}$ (Sanchez Goñi et al., *QSR*, 2008). How can the model-data difference in the magnitude of warming can affect the reliability of their model-based estimate of global mean temperature change?

- Lines 167-171 / lines 233-263: the authors should take into account new regional quantitative temperature records published in the last 20 years, since the Voelker et al. publication in 2002, and eventually run again the LOVECLIM model constrained by these new data. Thus, model-based global temperature estimations during the different D-O warming events will be more realistic. Consequently, the estimation of the feedback strengths of CO₂, CH₄ and N₂O will gain in reliability.

Minor correction

- Lines 444-445 : Please add the volume and ages of this reference.

Reviewer #2 (Remarks to the Author):

I have uploaded my review as a PDF.

Reviewer #3 (Remarks to the Author):

Review of "Past rapid warmings as a constraint on greenhouse-gas climate feedbacks" by Liu et al.

The paper estimates climate feedbacks to CO₂, CH₄, and N₂O forcing from proxies of the 19 Dansgaard-Oeschger events which are much constrained compared to previous feedback estimates from other past time-periods. To my knowledge this is a new, independent, and promising attempt.

The text reads well and the calculation seem correct from what I can assess. However, I cannot follow essential parts of the methods and am lost how to compare and interpret the feedback estimates to recent assessments of radiative feedbacks (Sherwood et al. 2020, AR6, ongoing discussion about quantification of feedbacks for modern/non-paleo settings).

Major:

The paper would have a higher impact if the numbers where related to the estimates in the recent assessment of climate sensitivity in Sherwood et al. 2020. This is the most thorough and up to date estimate of all lines of evidences, including paleo estimates. Expressing the feedbacks as gain here is not wrong, but make the comparison to the most current literature difficult (meaning, it leaves that job to the reader). As far as I understand, at least the CO₂ feedback is 1-1 comparable with feedback and ECS estimates in that report.

Similarly, the paper quantifies the “climate sensitivity parameter” (KW-1m-2) which is OK to use but much more uncommon than the climate feedback parameter (Wm-2K-1) — using AR6 notation would help comparing the numbers tremendously.

It stays unclear to me what timescales the feedbacks estimated here are referring to. “The D-O events provide an opportunity to quantify the warming-induced greenhouse-gas feedbacks to climate on a centennial timescale relevant to contemporary climate change” — this is vage. The title mentioned "rapid" (?) Are the estimates here comparable with the ones estimated within the next few decades, e.g. until 2100? What are centennial-scale feedbacks? How do they quantitatively relate to feedback estimates for decadal changes or equilibrium conditions?

The text doesn't hide but does also not highlight the fact that the forcing for the D-O events is in the order of magnitude of 10ppm, which is tiny compared to the ongoing forcing. 1900 through today and what's expected in the next decades. How does the small forcing relate to the centennial-time scale feedbacks?

In the D-O events, the CO₂ forcing acts on a much different timescale (before and after) than the temperature response (Fig.1). I think Fig.3 tries to explain why feedbacks can be still estimated in the classical forcing-feedback framework, in which the feedbacks always *follow* the forcing. However, I do not understand Fig.3, even with best intentions and after studying it several times. What is the bar length referring to? What is “effect of a feedback” referring to? Initially I thought the bars represent the feedback magnitude (but positive?) The words are not clear (e.g., climate induced concentration (of what?) feedback, carbon-concentration feedback vs. carbon-climate feedback). I don't understand what the up and down small arrow refer to. With “observed”, I guess, relates to proxies? Observations usually mean actually observable (which exist for radiative feedbacks), while proxies highlights their approximate nature.

There's a big discussion ongoing about feedbacks being sensitivity to the changes in underlying SST patten or ocean heat uptake (“pattern effect”). That effect makes “fast” feedbacks (water vapor, lapse rate, clouds, sea ice) change on centennial timescales. An adequate discussion of this (and potentially a quantitative comparison to e.g. “standard” feedbacks estimates in AR6, chapter 7) would be useful.

It stays unclear to me how the estimates for CO₂, CH₄, and N₂O are backed out from the single temperature response for each D-O event. In other words, I guess it is assumed that these feedbacks

are independent and add up to the overall D-O response (which is measured), but how, from the sum are the single feedbacks backed out?

In figure 2 the regression line is forced through the origin and in the text it is justly discussed whether or not that is required. Without forcing it through (0,0) there wouldn't be much or any correlation. is that a problem? What part of the findings depend on this regression?

Minor:

line 85: Probably not the authors mistake since AR6 is only out since August or so, but AR6 has new forcing estimates.

LOVECLIM does have fixed clouds, as far as I know. How does this assumption impact the results (potentially not much, because LOVECLIM is only used to translate the local to global temperatures?) — related: Could the uncertainties be relatively easily reduce through getting this global temperature estimates from other models as well?

Review of Liu et al., 2021, Past rapid warmings as a constraint on greenhouse-gas climate feedbacks

Summary

This manuscript attempts to constrain future greenhouse gas feedbacks operating in the earth system using past examples of abrupt climate change. The last glacial period is punctuated with a series of abrupt warmings and coolings observed in the Greenland ice cores. Coincident with these changes are roughly synchronous increases and decreases in CO₂, CH₄ and N₂O (allowing lags in the response). The analysis uses this relationship to quantify how much more CO₂, CH₄ and N₂O we might see in the future as global temperatures rise. The results rely on an intermediate complexity model (LOVECLIM) to transform Greenland temperature to estimates of global temperature. They then use statistical techniques to empirically estimate the feedback parameters.

Overall, I found the analysis to be overly simplistic and not very convincing. I have significant concerns about how a robust estimate of global temperature change is arrived at from effectively one single temperature record (Greenland) that is itself highly influenced by regional climate feedbacks and the redistribution of heat via the bipolar seesaw. I also found some of the logical steps difficult to follow at times, particularly the description of the maximum likelihood calculation. This may be evident in my review where I may be missing some crucial points, for which I apologize. None-the-less, I hope my comments will guide the authors to make the substantial changes needed before publication. I have also added a final section recommending a way for the paper to pivot towards a more defensible result.

In the current analysis the purely empirical constraint is based on some tenuous assumptions that I feel are not adequately justified.

- One model represents Greenland temperature well such that the model prediction can be linearly scaled to proxy data.
- Greenland temperature during DO events can be linearly related global temperature
- Global temperature can linearly related to greenhouse gas via feedbacks

Major Comments:

One model represents Greenland temperature well and can linearly transform proxy data

In reference to the DO events the paper states “The cause of these events is still under debate and several mechanisms have been invoked...” (line 52). With so many mechanisms, many of which are extremely difficult if not impossible to model, how confident are you that your one model can adequately transform Greenland temperature into global and regional patterns? Of course, this is not to ask you perfectly model a DO event but I feel one needs to acknowledge that this is a major limitation.

What if your model gets Greenland temperature roughly correct but for the wrong reasons? For example, sea-ice feedbacks play a huge role in driving the ice core d18O (Sime et al., 2019) (and also gas-derived temperatures). A strong regional sea-ice feedback would mean that relationship between Greenland temperatures and global temperature is very steep. A

weak regional sea-ice feedback would do the opposite to your relationship. How robust is this all the many other important feedbacks in our model?

For me, the model is not an accurate (enough) representation of the real world. The observed change in Greenland temperature varies from about -55C to -35C (Figure S2.9), yet the model only simulates a range of -36C to -24C. So there is both a severe underestimate of the variability (50%) and the mean values. Beyond this, the relationship between simulated Greenland temperature and global temperature is only about 15%.

The authors have done a good job propagating the large random errors that results from these relationships (assuming linearity). However, there is no attempt at quantify non-random errors. This would be very difficult and probably require additional models. I personally feel this leaves the current study to uncertain but I could be convinced if the paper was more upfront about these major shortcomings.

The paper would also benefit from a better description of the original model results. For example, the model was run with fixed with greenhouse gas concentrations. Therefore it would seem that the model is incapable of simulating the global temperature it is being employed to constrain. This seems like a major flaw to me. First off, why do you get a global temperature increase if the model is only redistributing heat? It would certainly be something interesting to follow up on. Run LOVECLIM forced with the greenhouse gas data. Run LOVECLIM with the freshwater forcing. Run them in combination. That should give you a better set of experiments to disentangle the global temperature response from Greenland signal as well as a set of experiments to investigate the model-consistent climate sensitivity, rather than employ an average of independent (ie potentially inconsistent) estimates (lines 111-113).

Greenland temperature can be linearly related global temperature

55-56 “The imprint of the D-O events is, nonetheless, reflected in large and globally synchronous changes in regional climates^{26–28} transmitted through the atmospheric circulation everywhere except Antarctica and surrounding regions.” This line is highlights the crux of the paper. Yet I think this is very simplistic view of D-O events and millennial-scale climate variability. For example, one just needs to look at figures S2.X which (if I understand correctly) shows the global temperature predictions of the model. Here it is clear that the impact of DO events is seen as a very restricted warming in the Arctic Ocean, Siberia and Greenland. The rest of the globe sees either negligible change or a cooling. From this, one could argue the exact opposite of the paper and suggest the Antarctic ice cores, rather than the Greenland ice cores are better representatives of global temperature.

Of course, the truth lies in between this and requires a much more nuanced approach than the one presented here. This pattern of millennial-scale change is quite well known, so I am surprised the authors took such a simplified approach and attempted to use Greenland temperatures to infer global temperatures and conclude that there is a global warming signal during DO events

Overall, additional analysis is needed to prove the statement (94-96) “LOVECLIM simulates the pattern of regional changes during individual D-O events during Marine Isotope Stage 3 well (Supplementary Figs 2.1 to 2.8), suggesting that our model-based estimate of global

mean temperature change is reliable.” Such plots are too qualitative to support this claim, please provide a quantitative analysis.

Global temperature can linearly related to greenhouse gas via feedbacks

My final point regards the notion that the variability in greenhouse gases can be so simply link to global temperatures. If I understand correctly, the statistical analysis treats variability amongst DO event as random errors (in the slope of GHGs-to-global temperature), but what about systematic errors? For example, what if all the CH₄ increase at a DO event is note driven by global temperature but is the results of some other process that happens to be correlated to Greenland temperature like changes in tropical hydroclimate? For CH₄, this is fairly well established. Overall, it was not clear to me how the analysis treats these potential for biases. Basically, it comes down the old question of correlation versus causation.

For CO₂, yes it has been argued that some of CO₂ increase could arise from the solubility feedback (Bauska et al., 2018; 2021) but this is isolated to the high-latitude North Atlantic. Moreover, it is also hypothesized that source of the CO₂ arises from a reinvigoration of AMOC (Marcott et al., 2014; Chen et al., 2015; Bauska et al., 2021, Skinner et al., 2021) which on balance, outweighs any CO₂ sinks from the Southern Ocean that are most likely activated. The resultant CO₂ increase is thus a fine balance of multi-factors, not all of which are directly linked to global temperature change. I realize that much of this evidence comes from the abrupt climate events of the last deglaciation. However, it's fairly well established that these are analogous to MIS3 but just better constrained by data.

This is a very different scenario to the last millennium, when at least a substantial portion of the temperature change is due to “external” forcing from solar and volcanic eruptions. Even here, though, signal is not global and the role of internal climate variability is heavily debated. That's why a regional approach to carbon-climate feedbacks has been used (Bauska et al., 2015).

For CH₄, it's quite clear the tropical and boreal hydroclimate is the primary driver of past emissions (see references below). Therefore, there is no straightforward link to global temperatures. There is also an issue, which I would argue is secondary, that the lifetime of CH₄ could be changing significantly across a DO event.

For N₂O, there is evidence that terrestrial temperatures are important. However, a substantial fraction of N₂O emissions come from marine sources (Schilt et al., 2014) which again, have no straightforward link to global temperature and more likely related to changes in AMOC (Jaccard et al., 2012). Again, the caveat that this from the deglaciation, but we know even less about individual DO events.

Also, there is the problem of using CO₂ concentrations to calculate the feedbacks rather than source histories. Imagine the CO₂ increase during an abrupt warming is due to the solubility feedback . If so, the rise in CO₂ would slightly lag the temperature increase but will eventually reach a new equilibrium. In this case, I see your method involving the before and after states being appropriate. But now imagine a scenario were the source of CO₂ is from an injection of terrestrial carbon. In this case, atmospheric CO₂ will peak and than slowly tail off over the timescales employed in the analysis and you could miss the transient response relevant to

future climate projections, and the apparent gain will be a constantly shifting target. The only way around this would be deconvolve source history of CO₂.

Note there is some data that limits the possibility of a purely terrestrial sources (Bauska et al., 2016; 2018) but there is a fair amount of literature claiming the opposite (e.g. Winterfeld, M. *et al.* Deglacial mobilization of pre-aged terrestrial carbon from degrading permafrost. *Nat. Commun.* **9**, 3666 (2018) and Köhler, P., Knorr, G. & Bard, E. Permafrost thawing as a possible source of abrupt carbon release at the onset of the Bølling/Allerød. *Nat Commun* **5**, 5520 (2014)).

CH₄ and N₂O are mostly immune to this as the source strength is roughly proportional to atmospheric concentration, assuming fixed lifetimes (should be mentioned).

A small note, but I think Figure 3 could use a lot more detail in the main text, rather than in the figure caption. These concepts might sit better in the introduction so the reader has bit stronger grasp of the concepts and complications in diagnosing feedbacks.

Future directions for analysis

I think the author's on the right track with analysis of climate-greenhouse feedbacks using all the interesting new ice core data. Following the discovering of the centennial-scale CO₂ events during the deglaciation, we have focused on understanding their mechanistic drivers for CO₂, CH₄ and N₂O. Yet there's an elephant in the room that needs pointing out. The fact is, the most of the changes are very small. Given the rapid and large warmings observed and Greenland (and more reasonably extended across much the high Arctic than the entire globe), why didn't we see major increases in CO₂? Perhaps it's the absence of any major feedbacks like permafrost dynamics, clathrates, peatland degradation? Of course, like the current study, caveats exists about the glacial period being analogue for the future.

I would recommend refocusing their question to ask "Is there an upper limit to high latitude Northern Hemisphere greenhouse gas feedbacks in response to abrupt warming?" Here the Greenland ice core records are much more suitable to answer this. This would also more closely align their results with the last millennium CC-feedback studies which are largely biased to the northern hemisphere. Note, Bauska et al., 2015 identified at possible positive feedback in the Arctic over the last millennium using a deconvolution of the land carbon history and Arctic temperatures, so there is possibility to further constrain this important feedback. Given all the complications listed above, I could imagine such a study could only rule out very large feedbacks rather than hone in a precise number that is directly comparable to the future, but that would still be an interesting result. Finally, a focus on the abrupt climate events of the last deglaciation where we have better global and regional temperature estimates (Shakun et al., 2012) and more climate modeling runs (e.g. <https://www.cgd.ucar.edu/ccr/TraCE/>) would lead to a more robust results.

Sincerely,
Thomas Bauska

Some ice core reference to delve deeper into the biogeochemistry of these events:

This is superior CH₄ dataset that could be directly related to the WAIS Divide CO₂ history. Also, I would have personally kept all the data sets on the superior WAIS Divide timescale (Buizert et al., 2014) rather than switch everything around to AICC2012.

Rhodes, R. H. *et al.* Enhanced tropical methane production in response to iceberg discharge in the North Atlantic. *Science* **348**, 1016–1019 (2015).

Broad-scale discussion of methane sources during glacial cycles

Bock, M. *et al.* Glacial/interglacial wetland, biomass burning, and geologic methane emissions constrained by dual stable isotopic CH₄ ice core records. *Proc. Natl. Acad. Sci.* **114**, E5778 (2017)

Discussion of CH₄ source histories across DO events:

Bock, M. *et al.* Hydrogen Isotopes Preclude Marine Hydrate CH₄ Emissions at the Onset of Dansgaard-Oeschger Events. *Science* **328**, 1686–1689 (2010).

Puts hard limits on geologic/clathrate emissions and support tropical sources

Dyonisius MN, et al., Old carbon reservoirs were not important in the deglacial methane budget. *Science*. 2020 Feb 21;367(6480):907-910. doi: 10.1126/science.aax0504.

Breaks down terrestrial and oceanic sources of N₂O:

Schilt, A. *et al.* Isotopic constraints on marine and terrestrial N₂O emissions during the last deglaciation. *Nature* **516**, 234–237 (2014).

Other References:

Buizert, C. *et al.* The WAIS Divide deep ice core WD2014 chronology - Part 1: Methane synchronization (68-31 ka BP) and the gas age-ice age difference. *Clim. Past* **11**, 153–173 (2015).

Chen, T. *et al.* Synchronous centennial abrupt events in the ocean and atmosphere during the last deglaciation. *Science* **349**, 1537 (2015).

Jaccard, S., Galbraith, E. Large climate-driven changes of oceanic oxygen concentrations during the last deglaciation. *Nature Geosci* **5**, 151–156 (2012).
<https://doi.org/10.1038/ngeo1352>

Köhler, P., Knorr, G. & Bard, E. Permafrost thawing as a possible source of abrupt carbon release at the onset of the Bølling/Allerød. *Nat Commun* **5**, 5520 (2014)

Shakun, J. D. *et al.* Global warming preceded by increasing carbon dioxide concentrations during the last deglaciation. *Nature* **484**, 49–54 (2012).

Sime, L. C., Hopcroft, P. O. & Rhodes, R. H. Impact of abrupt sea ice loss on Greenland water isotopes during the last glacial period. *Proc. Natl. Acad. Sci.* **116**, 4099–4104 (2019).

Skinner, L. C., Freeman, E., Hodell, D., Waelbroeck, C., Vazquez Riveiros, N., & Scrivner, A. E. (2021). Atlantic Ocean ventilation changes across the last deglaciation and their carbon cycle implications. *Paleoceanography and Paleoclimatology*, 36, e2020PA004074. <https://doi.org/10.1029/2020PA004074>

Winterfeld, M. *et al.* Deglacial mobilization of pre-aged terrestrial carbon from degrading permafrost. *Nat. Commun.* **9**, 3666 (2018)

Response to reviewers

We thank the reviewers for their comments on our manuscript and provide a point-by-point response below. The reviewers' comments are shown in *italics* and our responses in plain script, with revised text in blue. References in the comments to specific lines are to the original manuscript; line numbers in the responses refer to the new version of the manuscript.

Response to reviewer 1:

Major:

1. Lines 77-78: to reinforce their assumption that increase in global temperature leads the increase in GHG concentration it would be interesting to add in Figure 1 the curve of δD , a proxy for Antarctic temperature changes that are interpreted as global temperature variations.

Thank you very much for this suggestion. We have added the δD excess record from the EDC ice core, converted to the AICC 2012 timescale, to this plot. We chose the EDC core (rather than the EDML core) because although it has a slightly lower resolution (51 years versus 31 years), there is a sampling gap in the EDML record at ca 40 ka which is not present in the EDC core. We have added information about this record to Figure 1 and Supplementary Table 2. We note though that strictly speaking this is interpreted as temperature changes in the source area rather than global temperature (Markle et al., 2016). Nevertheless, it does clearly show the temperature changes associated with the D-O events.

2. Lines 79-80 and caption of Figure 1: why does the CO₂ concentration record only go back to 67 ka?

We used CO₂ concentrations from Bauska et al. (2021). This record has a high temporal resolution but only extends back to 67 ka. We considered merging this with other lower-resolution records but decided not to do so because this could cause unrealistic jumps in the record. In any case, there are only two events (DO-19, DO-20) that are excluded from the analyses of the CO₂ feedback because of this decision. We have modified the sentence in the main text to make the lengths of record clear:

Ice core records of the concentration of CO₂ during the period between 67 and 20 ka (Ref 30), and for CH₄ (Ref 31) and N₂O (Ref 32) during the period between 80 and 20 ka ...

We have modified the text describing these cores in the Methods section, as follows:

We used the ice-core records of atmospheric CO₂, CH₄ and N₂O concentrations detailed in Supplementary Table 2. The age models were converted to the Antarctic Ice Core Chronology 2012 (AICC2012) timescale³³ prior to analysis. The average resolution of the records on the AICC2012 timescale is 145 years for CO₂ (over the period 67 to 20 ka), 24 years for CH₄, and 67 years for N₂O over the period 80 – 20 ka ...

3. Lines 91-96 / lines 233-263: *The database of ref 37 is not updated and is mainly a compilation of sea surface temperature records with very few terrestrial ones. If the pattern of regional D-O warming events is relatively well reproduced by the LOVECLIM model, the magnitude of the warming is largely underestimated for the eastern part of the North Atlantic ocean and the western Mediterranean Sea. For instance, in the Alboran Sea, Gulf of Lyon and southwestern Iberian margin the SST increase during the D-O 8 and 12 is of ~9°C (Cacho et al., Paleoceanography, 1999 ; Pailler and Bard, Paleo 3, 2002 ; Català et al., Climate of the Past, 2019) while the simulated SST only show an increase of 2-4°C (Fig. S2.4, Fig. S2.8). In the northwestern Iberian margin and the Bay of Biscay, the estimated SST increase for D-O warming is even larger while in the model the increase remains at 2-4°C (Sanchez Goñi et al., QSR, 2008). How can the model-data difference in the magnitude of warming affect the reliability of their model-based estimate of global mean temperature change?*

Voelker et al. (2002) is, unfortunately, the most updated global synthesis currently available. Since LOVECLIM is a computationally efficient but low-resolution global climate model, it should not be expected to provide precise estimates for individual sites but rather to capture large-scale gradients in temperature. There are $64 \times 32 = 2048$ grids, so individual mismatches in a few grids is unlikely to affect the reliability of the model-based estimate of global mean temperature change too much.

However, we have made a more detailed data-model comparison for the Alboran Sea. Fig. 2 in Menviel et al. (2014) shows that the increase is ca 4 °C during DO 8 and 12 in this region, compatible with model estimates. Fig. 7d in Pailler and Bard (2002) shows the increase is ca 5 °C during DO 8 and ca 1 °C during DO 12, but these are low-resolution records. Fig. 3a in Cacho et al. (1999) shows the increase is 4–5 °C during DO 8 and 12. Fig. 4 in Català et al. (2019) also shows that the increase is 4–5 °C during DO 8 and 12.

We have expanded the text describing these experiments in the main text as follows:

There is limited quantitative information on regional temperature changes during the D-O warmings, but evaluation of the experiments against individual records^{37,39} as well as comparison with the global compilation of palaeoclimate data in Ref 40 shows that LOVECLIM simulates the pattern of regional changes during individual D-O events during Marine Isotope Stage 3 well (Supplementary Figs 2.1 to 2.8), suggesting that our model-based estimate of global mean temperature change is reliable (Supplementary Figs 2.9, 2.10).

We have also expanded the text in the Methods section to provide more details of the evaluation of these experiments as follows:

Analyses in the original paper³⁷ indicate that the simulations reproduce broadscale features of climate change during the D-O cycles, and there is a good match with quantitative estimates for specific D-O events (e.g. D-O 8) from the Iberian margin and western Mediterranean regions, where highly resolved SST records are available^{37,39}.

4. Lines 167-171 / lines 233-263: *the authors should take into account new regional quantitative temperature records published in the last 20 years, since the Voelker et al.*

publication in 2002, and eventually run again the LOVECLIM model constrained by these new data. Thus, model-based global temperature estimations during the different D-O warming events will be more realistic. Consequently, the estimation of the feedback strengths of CO₂, CH₄ and N₂O will gain in reliability.

The LOVECLIM simulations are not *constrained* by the data in Voelker et al. (2002). Rather, these data are used to evaluate the model performance. In addition to the use of the Voelker et al. data to evaluate the strength of the warming for specific time slices (DO-5 to DO-12) in this paper, the LOVECLIM model has been quantitatively evaluated against the speleothem ¹⁸O record (‰) from Sofular Cave, Turkey (Fleitmann et al., 2009) and the NGRIP temperature reconstruction (Huber et al., 2006) time series in Menviel et al (2014), and also compared to multiple records for DO-8 in Menviel et al (2020). Although this is a low-resolution model, its general performance is satisfactory.

As the reviewer correctly points out in the last comment, there are very few land-based temperature reconstructions in the Voelker et al (2002) compilation and it would be useful to incorporate more such reconstructions in our evaluation. We agree with the reviewer that it would be useful to generate an updated version of the Voelker et al. (2002) synthesis. Unfortunately, although there are many pollen records documenting D-O events (see e.g. Fletcher et al., 2010, for a summary of the European records), there are only a very few sites where quantitative temperature reconstructions have been made and the records are at high enough resolution to be identify specific D-O events reliably (Lake Ohrid, Lago di Monticchio, Villarquemado). The Lake Ohrid records provides estimates of mean annual temperature for Ognon I and II (D-O 19 and 20), which depending on the reconstruction method used (MAT or WAPLS) yields a warming of between 4 and 8 °C; unfortunately, these reconstructions cannot be used to evaluate the simulations, which do not encompass D-O 19 and 20. The clearest signals from Villarquemado are also for D-O 19 and D-O 20. Multiple D-O events are identified in the Lago di Monticchio record of changes in winter temperatures, and these appear to be consistent with the model results.

Minor:

5. *Lines 444-445: Please add the volume and pages of this reference.*

Thank you for pointing this out. This is a book chapter and we have now formatted this reference as shown.

Seager, R. & Battisti, D. Challenges to our understanding of the general circulation: Abrupt climate change. in *The Global Circulation of the Atmosphere* (eds. Schneider, T. & Sobel, A. H.) (2007).

Response to reviewer 2:

I found some of the logical steps difficult to follow at times, particularly the description of the maximum likelihood calculation.

We have revised the description of the logic and methods in the main text to try and make this clearer. We have modified the main text to make the reason for the choice clearer, as follows:

The feedback strength (in units of $\text{W m}^{-2} \text{K}^{-1}$) is the relationship between the radiative forcing brought about by the increases in CO_2 , CH_4 and N_2O and the increase in global mean temperature during D-O events (Fig. 2). A maximum likelihood method⁴¹ is used because it considers uncertainty of both the x- and y-variables, in contrast with ordinary least squares regression which assigns uncertainty only to the y-variable.

Major Comments:

1. One model represents Greenland temperature well and can linearly transform proxy data.

1.1 In reference to the DO events the paper states “The cause of these events is still under debate and several mechanisms have been invoked...” (line 52). With so many mechanisms, many of which are extremely difficult if not impossible to model, how confident are you that your one model can adequately transform Greenland temperature into global and regional patterns? Of course, this is not to ask you perfectly model a DO event but I feel one needs to acknowledge that this is a major limitation.

The specific mechanisms that give rise to D-O warmings are still under debate; the existence and spatial patterning of these warming events, however, is reasonably well known. Furthermore, even though the "ultimate" cause is unknown, most mechanisms involve the same proximate causes i.e. (a) the delayed reaction of the AMOC to the injection of a pulse of meltwater and (b) rapid atmospheric transmission of the resulting SST signal across a large part of the Earth's surface.

Our purpose in using a model is to translate measured Greenland temperature changes to an estimate of global temperature change, given the lack of sufficient evidence to construct a reliable estimate of the global temperature change during D-O events directly from observations. Thus it is sufficient that the model reproduces the spatial patterns of change reasonably well. Several models have mimicked the spatial patterns of D-O events using different types of forcing, and indeed some models produce these spontaneously without forcing – implying that the patterns are an inherent response to the reorganisation of atmospheric/ocean circulation. LOVECLIM is the only model that has set out to mimic specific D-O events by changing orbital parameters, atmospheric trace gas concentrations and ice sheet configuration appropriately through time, and adding meltwater pulses at the correct times required to trigger each event. Evaluation of the simulated patterns of temperature change, both in the original publications (see detailed comment below) and in our paper suggest that the simulated changes in global mean temperature are reasonably reliable. Since the LOVECLIM simulations only cover the interval between 50-30 ka, we use the relationship between Greenland temperature and global temperature over this interval to scale the Greenland

temperature estimates. The figure below shows this extension. The red line is LOVECLIM simulated global mean temperature anomaly to 30 ka; the black line is the inferred global mean temperature anomaly to 30 ka from Greenland ice core data using the method in our paper. We have added this figure to the Supplementary.

Having said this, we agree that it is worth discussing the caveats to our approach more fully, and have added text to address this and subsequent comments about the application of this model (see response below).

1.2 What if your model gets Greenland temperature roughly correct but for the wrong reasons? For example, sea-ice feedbacks play a huge role in driving the ice core $\delta^{18}O$ (Sime et al., 2019) (and also gas-derived temperatures). A strong regional sea-ice feedback would mean that relationship between Greenland temperatures and global temperature is very steep. A weak regional sea-ice feedback would do the opposite to your relationship. How robust is this all the many other important feedbacks in our model? For me, the model is not an accurate (enough) representation of the real world. The observed change in Greenland temperature varies from about $-55^{\circ}C$ to $-35^{\circ}C$ (Figure S2.9), yet the model only simulates a range of $-36^{\circ}C$ to $-24^{\circ}C$. So there is both a severe underestimate of the variability (50%) and the mean values. Beyond this, the relationship between simulated Greenland temperature and global temperature is only about 15%.

Chylek et al. (2005) used temperature records between 1975 and 2004, and found that the north-eastern part of Greenland is a strong indicator of global temperature change (correlation coefficient 0.91), and not influenced by the North Atlantic Oscillation. We assume that this relationship still existed in the last glacial period; this assumption is supported by the strong relationship between global mean temperature and north-eastern Greenland temperature in the model. Note that the value of 0.15 is the regression coefficient to transfer north-eastern Greenland temperature to global mean temperature. The correlation coefficient is 0.86.

Thus, the statistical relationship is close, but biased. We correct for this bias by finding the relationship between simulated and observed Greenland temperature as follows:

$$T_{\text{simulated Greenland}} = a_{1T} T_{\text{observed Greenland}} + a_{0T}$$

The relationship between simulated global mean temperature and simulated Greenland temperature is then given as:

$$T_{\text{simulated mean global}} = b_{1T} T_{\text{simulated Greenland}} + b_{0T}$$

Assuming

$$T_{\text{observed mean global}} = T_{\text{simulated mean global}}$$

we have

$$\begin{aligned} T_{\text{observed mean global}} &= b_{1T} (a_{1T} T_{\text{observed Greenland}} + a_{0T}) + b_{0T} \\ &= b_{1T} a_{1T} T_{\text{observed Greenland}} + b_{1T} a_{0T} + b_{0T} \end{aligned}$$

Since there is a relationship between observed global mean temperature and observed Greenland temperature, what matters now is not the value of a_{1T} , b_{1T} , a_{0T} , b_{0T} , but instead, of $b_{1T} a_{1T}$ and $b_{1T} a_{0T} + b_{0T}$. What we used for analysis is the change in global mean temperature, so $b_{1T} a_{0T} + b_{0T}$ also becomes unimportant. The figure in the last response shows that the black line ($b_{1T} a_{1T} \Delta T_{\text{observed Greenland}}$) matches reasonably well with the red line ($\Delta T_{\text{simulated mean global}}$). Therefore, $b_{1T} a_{1T}$ is on the right track for D-O events.

Sea-ice feedback is fast physical feedback, which can influence the value of $b_{1T} a_{1T}$ on a shorter timescale. Binning the records and simulations in 25-year bins however is expected to minimise the impact of fast physical feedbacks on the centennial-scale processes that are our focus.

1.3 The authors have done a good job propagating the large random errors that results from these relationships (assuming linearity). However, there is no attempt at quantify non-random errors. This would be very difficult and probably require additional models. I personally feel this leaves the current study to uncertain but I could be convinced if the paper was more upfront about these major shortcomings.

The largest non-random errors that potentially affect our calculations are those associated with the age modelling for the ice-core records, including the synchronization of ice and air ages. We agree that the contributions of non-random errors would not be easily quantified. As explained above, however, model biases are relatively unimportant since the model is only used to up-scale Greenland temperature. We have added some additional text about the age modelling in the revised manuscript as below:

The conversion to the AICC2012 chronology introduces additional uncertainties to those inherent in the original ice core age models, particularly for the earlier part of the record³³. Nevertheless, these are unlikely to have a significant effect given the method of determining the minimum and maximum response and thus of estimating the amplitude of change.

1.4 The paper would also benefit from a better description of the original model results. For example, the model was run with fixed greenhouse gas concentrations. Therefore, it would seem that the model is incapable of simulating the global temperature it is being employed to constrain. This seems like a major flaw to me.

For clarification, the LOVECLIM model was run using changing orbital parameters, changing atmospheric greenhouse gas concentrations, and changing ice sheet configuration between 50 and 30 ka; thus, the changes in forcing are appropriate for each of the simulated D-O events. This was stated in the Methods section of the original manuscript (lines 240 to 241). To make this clearer, we have revised the description of the experiments as follows:

The model was spun-up to equilibrium using an initial atmospheric CO₂ concentration of 207.5 ppm, orbital forcing appropriate for 50 ka BP and an estimate of the 50 ka BP ice-sheet orography and albedo obtained from an off-line ice-sheet model simulation⁶⁰. The transient run was initialised from this spin-up and run from 50 to 30 ka. Orbital, greenhouse gas, and ice-sheet forcings were updated continuously during the transient simulation; orbital parameters were derived following Ref 61, greenhouse gas concentrations were from ice core records, and the ice-sheet was from the off-line ice-sheet model simulation. In order to trigger D-O events, a time-series of freshwater inputs was derived by optimising freshwater fluxes such that simulated sea-surface temperature (SST) in the eastern subtropical North Atlantic were congruent with alkenone-based reconstructions of SST in that region.

The model is not being used to constrain the global temperature, but rather to up-scale the Greenland record to provide an estimate of global temperature changes.

We hope that our approach and logic are clearer in the revised manuscript.

1.5 First off, why do you get a global temperature increase if the model is only redistributing heat? It would certainly be something interesting to follow up on.

This happens because the model is not only redistributing heat spatially, but also redistributing heat between the surface and deep ocean. The global temperature increase in this paper means the *global surface air temperature* increase, which influences the biogeochemical cycles that in turn help to determine greenhouse gas concentrations.

1.6 Run LOVECLIM forced with the greenhouse gas data. Run LOVECLIM with the freshwater forcing. Run them in combination. That should give you a better set of experiments to disentangle the global temperature response from Greenland signal as well as a set of experiments to investigate the model-consistent climate sensitivity, rather than employ an average of independent (ie potentially inconsistent) estimates (lines 111-113).

It would be interesting to run multiple experiments to disentangle the impact of different forcings on the simulations, and indeed this is what is being proposed in the context of the planned PMIP work on D-O events. However, this is not the focus of our work. As we have explained above, the sole purpose of using the LOVECLIM simulations here is to up-scale the Greenland temperature records. We hope that this is now clearer in the revised manuscript.

In the original version of the paper, we used three independent estimates of the climate sensitivity in order to derive 95% confidence intervals for our estimates of gain. In the revised version, we have also used the ECS estimates provided by IPCC AR6 to calculate the gain. The propagated ECS in our paper is 3.23 ± 0.66 K (95% confidence interval); assuming that the combined likely range of the new estimates in IPCC AR6 can also be treated as a 95 % confidence interval, the ECS is 3.0 ± 0.5 K. As shown by the Tables below, the difference in ECS does not affect the estimates of the gain substantially, or affect our overall conclusions. The new tables also provide combined feedbacks of all the three greenhouse gases.

The table below shows feedback strength estimates (from the initial submission) using equations in IPCC AR5 to calculate radiative forcing brought about by greenhouse gases, with their 95 % confidence intervals.

	Feedback strength ($\text{W}\cdot\text{m}^{-2}\cdot\text{K}^{-1}$)	Gain using ECS in this paper	Gain using ECS in IPCC AR6
CO ₂	0.140 ± 0.019	0.122 ± 0.030	0.113 ± 0.024
CH ₄	0.085 ± 0.010	0.075 ± 0.018	0.069 ± 0.014
N ₂ O	0.082 ± 0.016	0.072 ± 0.020	0.067 ± 0.017
Combined	0.307 ± 0.027	0.268 ± 0.060	0.249 ± 0.047

In the revision we have also updated the radiative forcing equations to those in IPCC AR6. After this update, the resulting feedback strengths changed slightly. The Table below shows these revised estimates of feedback strength, and also the values of gain obtained using our ECS estimate and using the IPCC AR6 estimate:

	Feedback strength ($\text{W}\cdot\text{m}^{-2}\cdot\text{K}^{-1}$)	Gain using ECS in this paper	Gain using ECS in IPCC AR6
CO ₂	0.139 ± 0.019	0.121 ± 0.030	0.113 ± 0.024
CH ₄	0.096 ± 0.011	0.084 ± 0.020	0.078 ± 0.016
N ₂ O	0.077 ± 0.016	0.067 ± 0.020	0.062 ± 0.017
Combined	0.313 ± 0.027	0.274 ± 0.061	0.254 ± 0.048

In response to these various comments, we have modified the Discussion to emphasise the necessity (and implications) of using the LOVECLIM simulations as follows:

In contrast, the D-O warming events provide a strong constraint because the temperature changes, and the responses, are relatively large. Furthermore, replication over 19 events considerably reduces uncertainty compared to using a single event such as the LIA. Although we rely on a single model to derive a scaling between Greenland and global temperatures, the LOVECLIM model reproduces the major features of the global climate response during D-O cycles qualitatively and matches the limited number of quantitative climate-change estimates available^{37,39,40}. In the absence of a reliable observationally based reconstruction of global temperature, or of multiple simulations mimicking specific D-O events during the glacial, our approach provides a robust way to estimate greenhouse-gas climate feedbacks on centennial scales.

2. *Greenland temperature can be linearly related to global temperature.*

2.1 line 55-56: “The imprint of the D-O events is, nonetheless, reflected in large and globally synchronous changes in regional climates transmitted through the atmospheric circulation everywhere except Antarctica and surrounding regions.” This line highlights the crux of the paper. Yet I think this is very simplistic view of D-O events and millennial-scale climate variability. For example, one just needs to look at figures S2.X which (if I understand correctly) shows the global temperature predictions of the model. Here it is clear that the impact of DO events is seen as a very restricted warming in the Arctic Ocean, Siberia and Greenland. The rest of the globe sees either negligible change or a cooling. From this, one could argue the exact opposite of the paper and suggest the Antarctic ice cores, rather than the Greenland ice cores are better representatives of global temperature.

The Arctic Ocean, Siberia and Greenland do indeed show the largest warming. However, on closer inspection, these maps show that most of the global land surface also shows warming during D-O events. The warming is most pronounced in the extratropical regions because, as pointed out by Harrison and Sanchez Goñi (2010), the tropical response to D-O events is largely expressed through changes in moisture. Changes over land play an important role in the greenhouse gas feedbacks on centennial timescales.

We also assume a fixed ratio between global and global-land temperatures, as in Harrison et al. (2018). To avoid introducing extra error via a two-step process, we obtained the relationship between ΔT_{global} and $\Delta R_{\text{greenhouse gases}}$ in a single step:

$$\Delta R = \Delta R_{\text{land}} + \Delta R_{\text{ocean}}$$

$$\Delta R_{\text{land}} = g_{\text{land}} \Delta \bar{T}_{\text{land}}$$

$$\Delta R_{\text{ocean}} \approx 0$$

so

$$\Delta R = g_{\text{land}} \Delta \bar{T}_{\text{land}}$$

During D-O events, the land is consistently warming, but the ocean is warming in the north and cooling in the south.

However,

$$\Delta T_{\text{land}} = k \Delta \bar{T}$$

so

$$\Delta R = g_{\text{land}} k \Delta \bar{T} = g \Delta \bar{T}$$

$$g = \Delta R / \Delta \bar{T}$$

2.2 Of course, the truth lies in between this and requires a much more nuanced approach than the one presented here. This pattern of millennial-scale change is quite well known, so I am surprised the authors took such a simplified approach and attempted to use Greenland temperatures to infer global temperatures and conclude that there is a global warming signal during DO events.

We agree that the broadest-scale patterns of change during the D-O are relatively well known. However, there is very little quantitative information about the magnitude of these changes over land, and only a modest amount of information about changes in sea-surface temperatures outside the North Atlantic. This is why we have used simulations. During D-O events, most of the land was experiencing warming, the ocean was broadly warming in the north and cooling in the south, and the global mean temperature was increasing, as shown by the simulations. The feedbacks are mainly terrestrial; we assumed a fixed ratio between global and global-land temperature changes.

We inferred global mean temperature from the Greenland temperature record. Our approach is consistent with Chylek et al.'s (2005) finding of a strong correlation ($r = 0.91$) between recent (1975–2004) temperatures for north-eastern Greenland and global temperatures. We assume that this relationship still existed in the last glacial period. It is supported by the strong correlation ($r = 0.86$) between global mean temperature and Greenland temperature in the model.

In the revised manuscript, we have clarified the logic and approach we are using to make it clear why we take this up-scaling approach. Specifically, we have revised the text as follows:

An estimate of the global mean temperature change during each D-O warming event is required to calculate the associated feedback strength. There are too few quantitative reconstructions of temperature changes, especially over land, to be able to make reliable estimates. There is a strong correlation between temperature in north-eastern Greenland and global temperature over recent decades³⁶. There is a similarly highly significant relationship ($p < 0.001$, $R^2 = 0.75$) between north-eastern Greenland temperature and global mean temperature simulated by the LOVECLIM model in the suite of D-O hindcasts³⁷ covering the interval 50 – 30 ka. We therefore applied this relationship to derive the global mean temperature change from the NGRIP ice core record³⁸ between 80 – 20 ka. The resulting estimates of global mean temperature increase include a propagation of the uncertainties in this relationship.

2.3 Overall, additional analysis is needed to prove the statement (94-96) “LOVECLIM simulates the pattern of regional changes during individual D-O events during Marine Isotope Stage 3 well (Supplementary Figs 2.1 to 2.8), suggesting that our model-based estimate of global mean temperature change is reliable.” Such plots are too qualitative to support this claim, please provide a quantitative analysis.

The LOVECLIM model is constrained by the alkenone-based temperature reconstructions from the Iberian margin area. It has been compared quantitatively to the speleothem $\delta^{18}\text{O}$ record from Sofular Cave, Turkey and to the NGRIP temperature reconstructions in time series in Menviel et al (2014); to multiple records of DO-8 (Menviel et al., 2020); and, in this paper, to Voelker et al. (2002) for DO-5 through DO-12. We have detailed these comparisons in the revised manuscript. A more extensive quantitative analysis would require a larger data set of reconstructions that is currently not available. We have detailed the limitations in data availability, both in the response to other comments, and in the revised text (see e.g. lines 99-104, lines 192-199, lines 299-303).

3. Global temperature can be linearly related to greenhouse gas via feedbacks.

3.1 My final point regards the notion that the variability in greenhouse gases can be so simply link to global temperatures. If I understand correctly, the statistical analysis treats variability amongst DO event as random errors (in the slope of GHGs-to-global temperature), but what about systematic errors? For example, what if all the CH₄ increase at a DO event is not driven by global temperature but is the results of some other process that happens to be correlated to Greenland temperature like changes in tropical hydroclimate? For CH₄, this is fairly well established. Overall, it was not clear to me how the analysis treats these potential biases. Basically, it comes down the old question of correlation versus causation.

We do make the assumption that the causes of changes in emissions in one D-O event is the same as for all other D-O events. We recognize that the mechanisms involved in these feedbacks are not necessarily direct results of temperature changes. They are likely to involve changes in tropical hydroclimates, for example, that are correlated with changes in global temperature.

Such correlations are not accidental, however, nor do they represent a bias. We can draw a parallel with current understanding of the contemporary climate-CO₂ feedback, which is driven primarily by changes in tropical land climate (Cox et al., 2013; Wenzel et al., 2014) and is commonly quantified in the same manner as we have done here, in terms of global temperature – even though there is strong evidence that the feedback is primarily mediated by tropical hydroclimate (Humphrey et al., 2021, Nature). In other words, the whole idea of global feedback quantification rests on the (well-founded) assumption that global temperature changes do not occur alone, but are accompanied by changes in the spatial patterns of climate, including precipitation patterns. The details might differ among events, contributing to the variations that we treat as random in our analysis, but not destroying the strong observed relationships that form the basis of our estimates.

In light of this comment, we have added a paragraph in the Discussion section as follows:

We have assumed that there is a strong relationship between global temperature changes and greenhouse gas emissions during D-O warming events in order to estimate the climate feedback. Some of the changes in emissions may reflect change in hydroclimate, particularly in tropical regions²⁷, but we presume that such changes are also conditioned by changes in temperature and thus will be reflected in the global temperature record. Similarly, changes in the balance of marine versus terrestrial sources of greenhouse gas emissions, particularly CO₂, are influenced by the changes in global temperature. There is currently insufficient information to disentangle the regional sources of greenhouse gas emissions during the D-O events. However, the global feedback estimates obtained from analysis of the D-O events indicates that these feedbacks are non-negligible and poorly represented in current models.

3.2 For CO₂, yes it has been argued that some of CO₂ increase could arise from the solubility feedback (Bauska et al., 2018; 2021) but this is isolated to the high-latitude North Atlantic. Moreover, it is also hypothesized that source of the CO₂ arises from a

reinvigoration of AMOC (Marcott et al., 2014; Chen et al., 2015; Bauska et al., 2021, Skinner et al., 2021) which on balance, outweighs any CO₂ sinks from the Southern Ocean that are most likely activated. The resultant CO₂ increase is thus a fine balance of multi-factors, not all of which are directly linked to global temperature change. I realize that much of this evidence comes from the abrupt climate events of the last deglaciation. However, it's fairly well established that these are analogous to MIS3 but just better constrained by data. This is a very different scenario to the last millennium, when at least a substantial portion of the temperature change is due to "external" forcing from solar and volcanic eruptions. Even here, though, signal is not global and the role of internal climate variability is heavily debated. That's why a regional approach to carbon-climate feedbacks has been used (Bauska et al., 2015).

We agree that there may be several factors which influence natural CO₂ sources and sinks, that there may be changes in the balance of marine and terrestrial influences, and that CO₂ exchanges can also vary as a result of internal climate variability. However, we simply treat global temperature as an index of internal variability, linked to changes in hydroclimate and also to changes in both atmospheric and oceanic circulation. The analogy with the last millennium is unclear: it does not matter (for the purpose of calculating the feedback strength) whether the D-O events are forced or not.

3.3 For CH₄, it's quite clear the tropical and boreal hydroclimate is the primary driver of past emissions (see references below). Therefore, there is no straightforward link to global temperatures. There is also an issue, which I would argue is secondary, that the lifetime of CH₄ could be changing significantly across a D-O event.

Tropical hydroclimate is largely influenced by changes in atmospheric circulation and in particular the location of the ITCZ, which is influenced by changes in the pole-to-equator temperature gradient. The northward shift of the ITCZ during D-O warming events resulted in more moisture in the northern tropics, enhanced wetland activity there, and thus more CH₄ emissions. In boreal regions, changes in temperature and changes in moisture availability both play a role in determining CH₄ emissions. The changes in moisture availability are associated with changes in moisture advection, and again are a response to changes in temperature gradients.

Our calculations are based on concentrations, so any impact of climate change on atmospheric lifetimes is already implicitly included. In any case, the change in the atmospheric lifetime of CH₄ during D-O events is likely very small. Hopcroft et al. (2017) for example showed that the difference in atmospheric lifetime between the Last Glacial Maximum and the present was ca 2–8%; changes during D-O events presumably were not larger than this.

For completeness, we provide here a "back-of-the-envelope" comparison between the feedback with changing lifetime, and the feedback with fixed lifetime.

According to Hopcroft et al. (2017),

$$\frac{dB}{dt} = S - \frac{B}{\tau}$$

where B is the tropospheric CH₄ burden in Tg, S is the global CH₄ source (Tg CH₄ yr⁻¹) and τ is the tropospheric CH₄ lifetime in years. In steady state,

$$\frac{dB}{dt} = 0$$

so

$$S = \frac{B}{\tau}$$

The burden can be converted to concentration using a conversion factor k :

$$B = kc$$

so

$$S = \frac{kc}{\tau}$$

$$c = \frac{S}{k}\tau$$

When S and k are constant,

$$\frac{\Delta c}{c} = \frac{\Delta \tau}{\tau}$$

which means that the percentage increase in lifetime would result in the same percentage increase in concentration.

Hopcroft et al. (2017) indicated that the CH₄ lifetime during the LGM was 2–8% longer than that in pre-industrial time. Here we use CH₄ concentrations of 375 ppb for the LGM and 660 ppb for the pre-industrial period, as in Hopcroft et al. (2017). According to equation (3.5) in our Methods section, the radiative forcing brought about by CH₄ is a function of the minimum CH₄ concentration M_0 , the maximum CH₄ concentration M , and the mean N₂O concentration \bar{N} (\bar{M} is the average of M_0 and M ; \bar{N} is the average of N_0 and N):

$$\Delta R = f(M, M_0, \bar{N})$$

Assuming a mean N₂O concentration of 200 ppb (the influence of \bar{N} is small), the radiative forcing of CH₄ from LGM to pre-industrial level is

$$f(M = 660, M_0 = 375, \bar{N} = 200) = 0.257 \text{ Wm}^{-2}$$

The CH₄ lifetime is longer in cold conditions and shorter in warm conditions. If there were no lifetime change at LGM, in other words, if the lifetime were the same as the pre-industrial lifetime, the CH₄ concentration would be 375/(1+2%) to 375/(1+8%) ppb. The radiative forcing from LGM to the pre-industrial period would be between

$$f\left(M = 660, M_0 = \frac{375}{1 + 0.02}, \bar{N} = 200\right) = 0.265 \text{ Wm}^{-2}$$

and

$$f \left(M = 660, M_0 = \frac{375}{1 + 0.08}, \bar{N} = 200 \right) = 0.287 W m^{-2}$$

which is 3–12 % larger than the radiative forcing with changing lifetime.

3.4 For N₂O, there is evidence that terrestrial temperatures are important. However, a substantial fraction of N₂O emissions come from marine sources (Schilt et al., 2014) which again, have no straightforward link to global temperature and more likely related to changes in AMOC (Jaccard et al., 2012). Again, the caveat that this from the deglaciation, but we know even less about individual DO events.

Schilt et al. (2014) indeed showed that marine N₂O emissions contribute significantly to the total N₂O budget. However, they also noted that terrestrial emissions dominated atmospheric N₂O variation on centennial timescales. A biogeochemical model together with the data of Schilt et al. (2014), Joos et al. (2020, Biogeosciences) indicated a key role for temperature and precipitation changes over land in forcing large and rapid (decadal to centennial) N₂O increases during the deglaciation.

IPCC AR6 (Canadell et al., 2021) has provided quantitative estimates of the land N₂O feedback as $0.02 \pm 0.01 W m^{-2} K^{-1}$ (albeit with “low confidence”) and of the oceanic N₂O feedback as $-0.008 \pm 0.002 W m^{-2} K^{-1}$ (albeit based on “limited evidence”). Thus, as far as we know today, the total feedback due to both land and ocean effects is positive and dominated by the land, while the ocean feedback is smaller, and of opposite sign. The combined (land plus ocean) feedback strength for N₂O according to AR6 ($0.012 \pm 0.010 W m^{-2} K^{-1}$) is however considerably smaller than the value indicated by the D-O records.

In light of this new assessment, we have deleted the following sentence:

~~However, since these simulations do not take the ocean into consideration, the gains shown on Fig. 4 are lower than the D-O estimates of these gains.~~

and substituted the following text:

IPCC AR6 (Ref 46) estimated the land N₂O feedback as $0.02 \pm 0.01 W m^{-2} K^{-1}$ (with “low confidence”) and the oceanic N₂O feedback as $-0.008 \pm 0.002 W m^{-2} K^{-1}$ (based on “limited evidence”). Thus, AR6 indicates that the total N₂O feedback is positive and dominated by the land, while the ocean feedback is smaller and of opposite sign. The combined (land plus ocean) feedback strength for N₂O according to AR6 ($(0.02 - 0.008) \pm \sqrt{(0.01^2 + 0.002^2)} = 0.012 \pm 0.010 W m^{-2} K^{-1}$) however is considerably smaller than the value indicated by the D-O records.

3.5 Also, there is the problem of using CO₂ concentrations to calculate the feedbacks rather than source histories. Imagine the CO₂ increase during an abrupt warming is due to the solubility feedback. If so, the rise in CO₂ would slightly lag the temperature increase but will eventually reach a new equilibrium. In this case, I see your method involving the before and after states being appropriate. But now imagine a scenario where the source of CO₂ is from an injection of terrestrial carbon. In this case,

atmospheric CO₂ will peak and then slowly tail off over the timescales employed in the analysis and you could miss the transient response relevant to future climate projections, and the apparent gain will be a constantly shifting target.

The only way around this would be deconvolve source history of CO₂. Note there is some data that limits the possibility of a purely terrestrial sources (Bauska et al., 2016; 2018) but there is a fair amount of literature claiming the opposite (e.g. Winterfeld, M. et al. Deglacial mobilization of pre-aged terrestrial carbon from degrading permafrost. Nat. Commun. 9, 3666 (2018) and Köhler, P., Knorr, G. & Bard, E. Permafrost thawing as a possible source of abrupt carbon release at the onset of the Bølling/Allerød. Nat Commun 5, 5520 (2014)). CH₄ and N₂O are mostly immune to this as the source strength is roughly proportional to atmospheric concentration, assuming fixed lifetimes (should be mentioned).

This is an interesting thought experiment; but the ice core records make it clear that there were sustained increases in CO₂ concentration during D-O warming events, an observation that is not consistent with a single injection (see Figures in Supplementary). Thus, we argue that our before/after comparison is appropriate for CO₂, as well as for CH₄ and N₂O. We agree that the deconvolution of potential sources of CO₂ would be interesting, but we are constrained here by a lack of data for such an analysis.

3.6 A small note, but I think Figure 3 could use a lot more detail in the main text, rather than in the figure caption. These concepts might sit better in the introduction so the reader has bit stronger grasp of the concepts and complications in diagnosing feedbacks.

We have now removed the old Figure 3 (in response to reviewer 3) and carefully defined the different kinds of feedbacks. We specifically note that the simulated carbon-climate feedback consists of both a “pure” climate feedback and a climate-induced concentration feedback, and the feedback we quantified equals the carbon-climate feedback in simulations. We have provided a full algebraic explanation in Supplementary Materials, for completeness, but we hope that the main text is now less confusing than it was.

4. Future directions for analysis.

We thank the reviewer for these thoughts about potential future directions for analysis, although they are largely beyond the scope of the current paper.

4.1 I think the author's on the right track with analysis of climate-greenhouse feedbacks using all the interesting new ice core data. Following the discovering of the centennial-scale CO₂ events during the deglaciation, we have focused on understanding their mechanistic drivers for CO₂, CH₄ and N₂O. Yet there's an elephant in the room that needs pointing out. The fact is, the most of the changes are very small. Given the rapid and large warmings observed and Greenland (and more reasonably extended across much the high Arctic than the entire globe), why didn't we see major increases in CO₂? Perhaps it's the absence of any major feedbacks like permafrost dynamics, clathrates, peatland degradation?

The concentration changes are relatively small; however, the feedback induced by the small increase in CO₂ is larger than that from either CH₄ or N₂O. The three greenhouse gases combine to yield a gain of 0.213–0.335 (using AR6 equations and ECS). From this, we calculate that the global mean temperature change was 27–50 % larger than it would have been without greenhouse gas feedback.

The additional feedbacks mentioned by the reviewer are all, we presume, dependent on very much slower processes. They have been extensively discussed as potential causes of catastrophic (future) warming. We suggest that the evidence for this is thin; but this is a discussion we wish to avoid, at least in the present paper.

4.2 Of course, like the current study, caveats exist about the glacial period being analogue for the future. I would recommend refocusing their question to ask “Is there an upper limit to high latitude Northern Hemisphere greenhouse gas feedbacks in response to abrupt warming?” Here the Greenland ice core records are much more suitable to answer this. This would also more closely align their results with the last millennium CC-feedback studies which are largely biased to the northern hemisphere. Note, Bauska et al., 2015 identified a possible positive feedback in the Arctic over the last millennium using a deconvolution of the land carbon history and Arctic temperatures, so there is possibility to further constrain this important feedback. Given all the complications listed above, I could imagine such a study could only rule out very large feedbacks rather than hone in a precise number that is directly comparable to the future, but that would still be an interesting result.

We simply treat global mean temperature as an index of internal variability in the Earth system. We expect that such behaviour is spatially complex with different responses in different regions. We do not claim that the glacial is an analogue for the future. The rate and magnitude of warming are comparable; however, the cause is different. We explain that the abrupt warmings during D-O events are caused by oceanic re-organisations (from one cause or another) and that the changes in greenhouse gases are caused by the changes in temperature rather than vice versa (lines 58-60 in original manuscript). Our point here is that these warming events provide a way of quantifying the climate feedbacks from the observational record, and doing so over multiple events.

4.3 Finally, a focus on the abrupt climate events of the last deglaciation where we have better global and regional temperature estimates (Shakun et al., 2012) and more climate modelling runs (e.g. <https://www.cgd.ucar.edu/ccr/TraCE/>) would lead to a more robust results.

We agree that it would be useful to examine feedbacks associated with the abrupt climate changes during the last deglaciation. However, the Shakun et al (2012) temperature reconstruction is severely biased towards coastal ocean regions and has been shown to be unrepresentative (see Marsicek et al., 2018). The TRACE21k model does indeed provide results that mimic the abrupt changes during the deglaciation, although the longitudinal resolution is relatively coarse (3.6°). A number of other modelling groups are currently running deglacial simulations as part of the Palaeoclimate Modelling Intercomparison Project and these could be a useful basis for further analysis when the outputs become available. However, the main problem with using the deglaciation for our purpose is the limited number of these abrupt warmings, with only the start of the Bølling-Allerød and the end of the Younger Dryas as targets.

Thus, while analyses of these events would certainly complement the analyses of D-O warmings, it is unclear whether they would provide a stronger constraint.

What would be useful however is to combine these deglaciation simulations (when available) with planned simulations of specific D-O events during Marine Isotope Stage 3 to derive a larger range of estimates of the change in global mean temperature. This kind of analysis also requires a concerted effort to derive reliable quantitative estimates of regional climate changes during both periods with a sufficiently representative coverage to be able to derive reliable estimates of global temperature changes.

4.4 This is superior CH₄ dataset that could be directly related to the WAIS Divide CO₂ history. Also, I would have personally kept all the data sets on the superior WAIS Divide timescale (Buizert et al., 2014) rather than switch everything around to AICC2012.

Rhodes, R. H. et al. Enhanced tropical methane production in response to iceberg discharge in the North Atlantic. Science 348, 1016–1019 (2015).

Thank you very much for this suggestion! The record from the Rhodes et al. paper has extremely high temporal resolution because it was measured continuously. However, since we need to have a mean value for N₂O in the calculation of CH₄ radiative forcing (see equation 3.5), we prefer to derive both N₂O and CH₄ from the Greenland records. The choice of the AICC2012 timescale was based on advice from Eric Wolff. In any case, the choice of timescale would have minimal impact on the regressions.

Response to reviewer 3:

1. I cannot follow essential parts of the methods and am lost how to compare and interpret the feedback estimates to recent assessments of radiative feedbacks (Sherwood et al. 2020, AR6, ongoing discussion about quantification of feedbacks for modern/non-paleo settings).

We have now revised the Methods section and we hope that our approach is now clearer. As detailed in our response below, we have revised the manuscript to make use of the radiative forcing equations provided by Etminan et al (2016) and used in the AR6. We have preserved the ECS estimates used in the original manuscript which derived from Cox et al. (2018), Sherwood et al. (2020) and Tierney et al. (2020) because these allowed us to derive 95% confidence intervals explicitly. However, we have also used the likely range of values provided in the AR6, on the assumption that this range can be treated as equivalent to a 95% confidence interval. We provide the feedback strength (in W m⁻² K⁻¹) for each greenhouse gas using AR6 radiative forcing equations and the gains estimated from the two different approaches to derive the ECS in the revised text. We have also added a comparison of feedback strengths (rather than gains) to the Supplementary, and we refer to this in the revised text.

Major:

2. The paper would have a higher impact if the numbers were related to the estimates in the recent assessment of climate sensitivity in Sherwood et al. 2020. This is the most thorough and up to date estimate of all lines of evidences, including paleo estimates.

Expressing the feedbacks as gain here is not wrong, but make the comparison to the most current literature difficult (meaning, it leaves that job to the reader). As far as I understand, at least the CO₂ feedback is 1-1 comparable with feedback and ECS estimates in that report. Similarly, the paper quantifies the “climate sensitivity parameter” ($KW^{-1}m^{-2}$) which is OK to use but much more uncommon than the climate feedback parameter ($Wm^{-2}K^{-1}$) — using AR6 notation would help comparing the numbers tremendously.

We express the feedbacks as gain because this measure has a simple intuitive interpretation. However, we agree that it is helpful also to provide estimated feedback strengths ($W m^{-2} K^{-1}$) for comparison with the literature. Indeed, we gave estimates of the feedback strength in the original text (lines 100-101) and in Table 1 of the original manuscript. In response to a subsequent comment, we have updated this Table to include ECS estimates from AR6 (the impact is minor, see below), on the assumption that the likely range of the combined assessment can be taken as equivalent to the 95% confidence interval. We still give both the feedback strength and the gain (see lines 109-111 and lines 121-128 in the revised text). We have now added a comparison with other estimates in terms of feedback strengths (rather than gains) to the Supplementary, to facilitate comparisons.

Sherwood et al. (2020) quantified only the fast physical feedbacks, and not the CO₂ feedback. The reference system for fast physical feedbacks is the black body temperature, whereas the reference system for greenhouse-gas feedbacks is the combination of black body temperature with fast physical feedbacks. Thus, the ECS in Sherwood et al. (2020) is equivalent to the ECS of the reference system in our paper. We used the Sherwood et al. (2020) estimate as one of the three sources of information about ECS: i.e. the estimates of 2.2 to 3.4 K (66 % CI) (Cox *et al.* 2018), 2.3 to 4.7 K (95% CI) (Sherwood *et al.* 2020) and 2.4 to 4.5 K (95% CI) (Tierney *et al.* 2020). The propagated ECS is 3.23 ± 0.66 K (95% confidence interval). The ECS using the AR6 likely range is 3.0 ± 0.5 K.

3. It stays unclear to me what timescales the feedbacks estimated here are referring to. “The D-O events provide an opportunity to quantify the warming-induced greenhouse-gas feedbacks to climate on a centennial timescale relevant to contemporary climate change” — this is vague. The title mentioned “rapid” (?) Are the estimates here comparable with the ones estimated within the next few decades, e.g. until 2100? What are centennial-scale feedbacks? How do they quantitatively relate to feedback estimates for decadal changes or equilibrium conditions?

The D-O warming is indeed rapid, and as we stated in the original manuscript (lines 45-46). The rate of warming during D-O events is comparable to the rate of $0.041^{\circ}C$ /year in north-eastern Greenland between 1975 and 2004 (Chylek et al., 2005) and also comparable to the rate of change for mid-value projections of the 21st century according to the IPCC AR6 report. The time taken for the greenhouse gas changes (minimum to maximum values) during individual D-O events varies, but is of the order of centuries (see Table below):

DO	CO ₂ (year)	CH ₄ (year)	N ₂ O (year)	Greenland temperature (year)	Greenland temperature change rate (°C/year)
2	225	350	300	225	0.035
3	450	150	300	250	0.053
4	325	275	100	225	0.055
5	425	350	400	250	0.049
6	175	375	200	250	0.035
7	375	400	300	175	0.063
8	250	300	300	225	0.045
9	300	250	300	125	0.050
10	375	275	300	200	0.068
11	500	325	600	225	0.055
12	425	325	500	250	0.050
13	225	300	500	175	0.049
14	425	375	400	200	0.063
15	425	175	500	200	0.049
16	250	325	100	150	0.068
17	250	275	600	225	0.057
18	225	350	400	250	0.041
19	/	250	400	225	0.061
20	/	350	400	200	0.061

Since it might be helpful for the reader, we have added this summary Table to the Supplementary. We have added the reference (Chylek et al., 2005) to this Table.

4. The text doesn't hide but does also not highlight the fact that the forcing for the D-O events is in the order of magnitude of 10 ppm, which is tiny compared to the ongoing forcing. 1900 through today and what's expected in the next decades. How does the small forcing relate to the centennial-time scale feedbacks?

The small change in CO₂ observed during D-O events is *not the forcing* for those events. Our analysis for CO₂ rests on the (reasonable) assumption that it is a consequence, rather than a cause, of the warming. This does not mean that the resulting feedback is trivial, however.

*5. In the D-O events, the CO₂ forcing acts on a much different timescale (before and after) than the temperature response (Fig.1). I think Fig.3 tries to explain why feedbacks can be still estimated in the classical forcing-feedback framework, in which the feedbacks always *follow* the forcing. However, I do not understand Fig.3, even with best intentions and after studying it several times. What is the bar length referring to? What is "effect of a feedback" referring to? Initially I thought the bars represent the feedback magnitude (but positive?) The words are not clear (e.g., climate induced concentration (of what?) feedback, carbon-concentration feedback vs. carbon-climate feedback). I don't understand what the up and down small arrow refer to. With "observed", I guess, relates to proxies? Observations usually mean actually observable (which exist for radiative feedbacks), while proxies highlight their approximate nature.*

In response to this comment, we have eliminated this figure, and clarified the text on different types of feedback. We specifically note that the simulated carbon-climate

feedback consists of both a “pure” climate feedback and a climate-induced concentration feedback, and the feedback we quantified equals the carbon-climate feedback in simulations. We have provided a full algebraic explanation in Supplementary Materials, for completeness, but we hope that the main text is now less confusing than it was. The relevant (revised) section is quoted here:

Model-based feedback estimates have been derived from simulations of the response to anthropogenic emissions, and separate the carbon-concentration feedback and the carbon-climate feedback⁸. Changes in the atmospheric carbon concentration caused by emissions are buffered by the land and ocean uptake by the carbon-concentration feedback (a negative feedback); the amount of carbon these sinks can absorb is reduced by the carbon-climate feedback (a positive feedback)⁸. In the present-day context, anthropogenic CO₂ emissions are the main driver of changes in the carbon cycle and warming is the response of the emissions; in the D-O context, warming is the main driver and changes in the atmospheric CO₂ is the response. The feedback we quantified using D-O warmings equals the carbon-climate feedback defined in Ref 8. See Supplementary Materials for a more detailed analysis.

6. There’s a big discussion ongoing about feedbacks being sensitivity to the changes in underlying SST pattern or ocean heat uptake (“pattern effect”). That effect makes “fast” feedbacks (water vapor, lapse rate, clouds, sea ice) change on centennial timescales. An adequate discussion of this (and potentially a quantitative comparison to e.g. “standard” feedbacks estimates in AR6, chapter 7) would be useful.

Dong et al.’s recent paper in the *Journal of Climate* has sparked an interesting discussion about the role of changes in SSTs and ocean heat uptake on climate sensitivity. However, our understanding is that the higher climate sensitivity of the CMIP6 models, compared to CMIP5, is not a reflection of this, but rather of differences in the strength of the cloud feedback. Furthermore, the CMIP6 models apparently have a stronger response on short timescales than on the longer timescales that would reflect the pattern effect. In the revised manuscript we have added estimates of ECS and the climate sensitivity parameter from AR6. Both estimates are smaller than our previous calculations, but the difference is slight.

7. It stays unclear to me how the estimates for CO₂, CH₄, and N₂O are backed out from the single temperature response for each D-O event. In other words, I guess it is assumed that these feedbacks are independent and add up to the overall D-O response (which is measured), but how, from the sum are the single feedbacks backed out?

The response that is measured is the response of each GHG separately, so it makes sense to calculate the radiative forcing and feedbacks separately for each GHG. The overall GHG feedback is not measured, but it is indeed calculated by summing the radiative forcing of the three GHGs. We have now added a panel showing the combined forcing to Figure 2, and we have revised the Methods section to clarify how these results were obtained.

8. In figure 2 the regression line is forced through the origin and in the text it is justly discussed whether or not that is required. Without forcing it through (0,0) there wouldn’t be much or any correlation. is that a problem? What part of the findings depend on this regression?

These regressions play a central role in our calculations, as our estimated feedback strengths and gains depend proportionally on the fitted slopes. ΔR is assumed proportional to ΔT (Knutti & Hegerl 2008; Roe 2009):

$$\Delta R = c\Delta T$$

therefore, the regression line between ΔR and ΔT *should* be forced through the origin. A regression with an intercept in this context would not make physical sense.

However, although this calculation does not formally require there to be a significant correlation between ΔR and ΔT , the correlations are 0.24 for CO₂, 0.47 for CH₄, 0.42 for N₂O and 0.67 for the combined forcing. Except for CO₂, all of these are significant.

Minor:

9. line 85: Probably not the authors mistake since AR6 is only out since August or so, but AR6 has new forcing estimates.

Indeed, the AR6 report came out several months after we had completed our analyses and submitted this paper. However, since it is important to include the latest estimates, we have now included the AR6 forcing estimates in our paper. One minor difficulty here is that whereas the 95% confidence intervals were given for the ECS in AR5, AR6 only gives a likely range and we have had to assume this can be interpreted as equivalent to the 95 % confidence interval.

The table below shows feedback strength estimates (from the initial submission) using equations in IPCC AR5 to calculate radiative forcing brought about by greenhouse gases, with their 95 % confidence intervals. ECS in this paper is 3.23 ± 0.66 K (95% confidence interval), ECS in IPCC AR6 is 3.0 ± 0.5 K:

	Feedback strength (W m ⁻² K ⁻¹)	Gain using ECS in this paper	Gain using ECS in IPCC AR6
CO ₂	0.140 ± 0.019	0.122 ± 0.030	0.113 ± 0.024
CH ₄	0.085 ± 0.010	0.075 ± 0.018	0.069 ± 0.014
N ₂ O	0.082 ± 0.016	0.072 ± 0.020	0.067 ± 0.017
Combined	0.307 ± 0.027	0.268 ± 0.060	0.249 ± 0.047

In the revised manuscript, we have also used the IPCC AR6 equations to calculate greenhouse-gas radiative forcings:

	Feedback strength (W m ⁻² K ⁻¹)	Gain using ECS in this paper	Gain using ECS in IPCC AR6
CO ₂	0.139 ± 0.019	0.121 ± 0.030	0.113 ± 0.024
CH ₄	0.096 ± 0.011	0.084 ± 0.020	0.078 ± 0.016
N ₂ O	0.077 ± 0.016	0.067 ± 0.020	0.062 ± 0.017
Combined	0.313 ± 0.027	0.274 ± 0.061	0.254 ± 0.048

The change to the CO₂ feedback is trivial; the CH₄ feedback becomes slightly larger and the N₂O feedback slightly smaller. However, these changes do not impact the

conclusions of our study, and the analysis still provides a better constraint on the magnitude of these feedbacks than existing studies.

10. LOVECLIM does have fixed clouds, as far as I know. How does this assumption impact the results (potentially not much, because LOVECLIM is only used to translate the local to global temperatures?)

It is true that the LOVECLIM simulations were run with fixed cloud cover in these hindcast experiments. Studies examining the impact of using fixed clouds (e.g. Zhang et al. 2010, *J. Clim*) suggest that changes in cloud cover accentuate the temperature changes: it gets colder in NH, particularly in the North Atlantic region, but gets warmer in SH. The enhanced NH cooling and SH warming can compensate each other so that the impact on global temp is small. Since we are only using the LOVECLIM simulations to convert local Greenland to global mean temperature, we agree with the reviewer that the impact of using fixed clouds here is presumably negligible.

11. Could the uncertainties be relatively easily reduced through getting this global temperature estimates from other models as well?

LOVECLIM is the *only* model used so far to have been used to hindcast specific D-O events. Several models have been used to simulate D-O-like events based on adding freshwater forcing to a glacial-state simulation, but these maintain constant boundary conditions specific to the Last Glacial Maximum, and thus do not provide the range of different D-O responses simulated by LOVECLIM. There is an initiative within the Palaeoclimate Modelling Intercomparison Project to investigate D-O-like events both in response to idealised forcing and explicitly to forcings appropriate to D-O events during Marine Isotope Stage 3. It would be useful to compare our results using LOVECLIM with simulations by other models, when these become available.

Additional changes to update the text to IPCC AR6

Consideration of CMIP6 simulations of carbon-climate feedback:

Model estimates of the carbon-climate feedback based on simulations from the Coupled Climate Carbon Cycle Model Intercomparison (C⁴MIP)⁸ show considerable variability, with estimates of the gain ranging from 0.04 to 0.31 (Fig. 3; see Supplementary Fig. 3.1 for comparison of feedback strengths). The range is somewhat reduced in models from the fifth and sixth phases of the Coupled Model Intercomparison Project (CMIP5¹⁰, CMIP6⁴⁵): 0.03 to 0.18 in CMIP5 and - 0.002 to 0.18 in CMIP6. Our estimate of the CO₂ gain derived from the D-O warming events is not consistent with high-end estimates from C⁴MIP, nor with low-end estimates from C⁴MIP, CMIP5 and CMIP6.

Consideration of estimated CH₄-climate feedbacks in AR6:

IPCC AR6 (Ref 46) estimated the CH₄-climate feedback due to the effect of temperature on methanogenesis in wetlands as $0.03 \pm 0.01 \text{ W m}^{-2} \text{ K}^{-1}$ (1 standard deviation, based on “limited evidence”) and an additional, highly uncertain feedback of 0.01 (0.003 to 0.04, 5th to 95th percentile range, also based on “limited evidence”) W

$\text{m}^{-2} \text{K}^{-1}$ due to permafrost thaw. Our results suggest that the CH_4 -climate feedback is larger than was assessed by AR6.

Consideration of estimated N_2O -climate feedbacks in AR6:

IPCC AR6 (Ref 46) estimated the land N_2O feedback as $0.02 \pm 0.01 \text{ W m}^{-2} \text{ K}^{-1}$ (with “low confidence”) and of the oceanic N_2O feedback as $-0.008 \pm 0.002 \text{ W m}^{-2} \text{ K}^{-1}$ (based on “limited evidence”). Thus, AR6 indicates that the total N_2O feedback is positive and dominated by the land, while the ocean feedback is smaller and of opposite sign. The combined (land plus ocean) feedback strength for N_2O according to AR6 $((0.02 - 0.008) \pm \sqrt{(0.01^2 + 0.002^2)}) = 0.012 \pm 0.010 \text{ W m}^{-2} \text{ K}^{-1}$ however is considerably smaller than the value indicated by the D-O records.

14th Apr 22

Dear Ms Liu,

Please allow me to apologise for the long delay in sending a decision on your manuscript titled "Past rapid warmings as a constraint on greenhouse-gas climate feedbacks". It has now been seen again by Reviewer #1 and also by a new Reviewer #4. Unfortunately the previous reviewers #2 and #3 were unable to provide further reports. I include the comments of Reviewers #1 and #4 at the end of this message. They find your work of interest, but Reviewer #4 raises some important points. We remain interested in the possibility of publishing your study in *Communications Earth & Environment*, but would like to consider your responses to these concerns and assess a revised manuscript before we make a final decision on publication.

We therefore invite you to revise and resubmit your manuscript, along with a point-by-point response that takes into account the points raised. Please highlight all changes in the manuscript text file.

In particular, please ensure that in the revised manuscript you clearly outline the sources of systematic error and quantify the potential effects of model bias.

Please use the following link to submit your revised manuscript, point-by-point response to the referees' comments (which should be in a separate document to any cover letter) and the completed checklist:

[link redacted]

We hope to receive your revised paper within six weeks; please let us know if you aren't able to submit it within this time so that we can discuss how best to proceed. If we don't hear from you, and the revision process takes significantly longer, we may close your file. In this event, we will still be happy to reconsider your paper at a later date, as long as nothing similar has been accepted for publication at *Communications Earth & Environment* or published elsewhere in the meantime.

We understand that due to the current global situation, the time required for revision may be longer than usual. We would appreciate it if you could keep us informed about an estimated timescale for resubmission, to facilitate our planning. Of course, if you are unable to estimate, we are happy to accommodate necessary extensions nevertheless.

Please do not hesitate to contact me if you have any questions or would like to discuss these revisions further. We look forward to seeing the revised manuscript and thank you for the opportunity to review your work.

Best regards,

Joe Aslin

Senior Editor,
Communications Earth & Environment
<https://www.nature.com/commsenv/>
Twitter: @CommsEarth

EDITORIAL POLICIES AND FORMATTING

Editorial Policy: [Policy requirements](https://www.nature.com/documents/nr-editorial-policy-checklist.zip)

Furthermore, please align your manuscript with our format requirements, which are summarized on the following checklist:

[Communications Earth & Environment formatting checklist](https://www.nature.com/documents/commsj-phys-style-formatting-checklist-article.pdf)

and also in our style and formatting guide [Communications Earth & Environment formatting guide](https://www.nature.com/documents/commsj-phys-style-formatting-guide-accept.pdf) .

***** DATA:** Communications Earth & Environment endorses the principles of the Enabling FAIR data project (<http://www.copdess.org/enabling-fair-data-project/>). We ask authors to make the data that support their conclusions available in permanent, publically accessible data repositories. (Please contact the editor if you are unable to make your data available).

All Communications Earth & Environment manuscripts must include a section titled "Data Availability" at the end of the Methods section or main text (if no Methods). More information on this policy, is available at <http://www.nature.com/authors/policies/data/data-availability-statements-data-citations.pdf>.

DATA SOURCES: All new data associated with the paper should be placed in a persistent repository where they can be freely and enduringly accessed. We recommend submitting the data to discipline-specific, community-recognized repositories, where possible and a list of recommended repositories

is provided at <http://www.nature.com/sdata/policies/repositories>.

If a community resource is unavailable, data can be submitted to generalist repositories such as [figshare](https://figshare.com/) or [Dryad Digital Repository](http://datadryad.org/). Please provide a unique identifier for the data (for example a DOI or a permanent URL) in the data availability statement, if possible. If the repository does not provide identifiers, we encourage authors to supply the search terms that will return the data. For data that have been obtained from publically available sources, please provide a URL and the specific data product name in the data availability statement. Data with a DOI should be further cited in the methods reference section.

REVIEWER COMMENTS:

Reviewer #1 (Remarks to the Author):

Liu et al. have satisfactorily taken into account my major and minor criticisms, and modified the manuscript accordingly. Moreover, the revised manuscript reads now clearer. I have only one comment: in lines 119-120 the authors should add the abbreviation (ECS) after "estimate of climate sensitivity". I strongly recommend the publication of the revised manuscript.

Reviewer #4 (Remarks to the Author):

I was asked to review this resubmission in light of Reviewer 2's (R2's) concerns about the use of model constraints.

The design of the study is ambitious and makes a number of assumptions to reach its conclusions. A main request of R2 was to make these assumptions more explicit throughout the paper. While the authors have clarified some important points and improved readability, I think that they have tried to argue away some sources of uncertainty that are in fact fundamental. Straightforward statements of the shortcomings of these assumptions are still needed following the suggestions of R2, including a discussion of implications for the accuracy of the confidence intervals given, which are an essential result. Specifically, numbered as in the rebuttal document:

1.1 The authors argue that "...it is sufficient that the model reproduces the spatial patterns of change reasonably well." I think it is currently somewhat incongruent within the paper to have an emphasis on uncertainty propagation paired with a less rigorous examination of a model used for a critical step. What does "reasonable" mean here? How might errors in the model or the assumption that the covariance of D-O events looks like that of AMOC hosing in a low-resolution model translate into errors in the final estimate? While it's beyond the scope of the paper to quantify all of these

possibilities, the authors must at least discuss them since it seems like a substantial shortcoming.

1.3 Related to 1.1, the effects of model biases have not been quantified, so I don't think the authors can argue that they are small. "Bias" in this instance extend beyond a time-mean model bias to errors in regression coefficients, which can be large (see e.g. Amrhein et al. 2020 for a discussion of quantifying structural uncertainties arising from model regression relationships). Please discuss the implications of regression errors linking Greenland ice cores and global mean temperatures during D-O events for your results.

2.2 The authors argue multiple times that a correlation observed over modern times between Greenland and global mean temperature points to a similar relationship for D-O events. However, this logic and its shortcomings needs to be explored more. Climate variability has more than one spatial mode, and it's not obvious why a regression coefficient computed over recent decades (in the presence of anthropogenic warming) should argue for a similar relationship for D-O events.

Moreover, I find that the statement in the revised manuscript that "The resulting estimates of global mean temperature increase include a propagation of the uncertainties in this relationship" is not correct because it neglects these issues of model regression error.

In summary, the authors need to lay out these sources of systematic error and their impacts on results clearly. In addition, I would strongly encourage the authors to provide an estimate of this uncertainty in their confidence intervals, which would strengthen the paper. One approach could be to take Greenland-global regression coefficients from LOVECLIM, use them to reconstruct global mean temperature from a Greenland location in another model, and compare that reconstruction to the true GMT value in that model. The resulting estimate of the effects of regression error could be propagated to the final results.

Finally, I was also personally confused by the ability of the regression procedure to distinguish among feedbacks from the three GHGs considered. Given that the history of GHGs is not uncorrelated in time, how are feedbacks partitioned among the gases? Is the problem well-posed? It would help to show sensitivity to standard errors of radiative forcing (which can be important for partitioning solution variance in least-squares problems) and other relevant parameters to argue for robustness.

Response to reviewers

We thank the reviewers for their comments on our manuscript and provide a point-by-point response below. The reviewers' comments are shown in *italics* and our responses in plain script, with revised text in **blue**. References in the comments to specific lines are to the original manuscript; line numbers in the responses refer to the new version of the manuscript.

Reviewer 1:

Liu et al. have satisfactorily taken into account my major and minor criticisms, and modified the manuscript accordingly. Moreover, the revised manuscript reads now clearer. I have only one comment: in lines 119-120 the authors should add the abbreviation (ECS) after "estimate of climate sensitivity". I strongly recommend the publication of the revised manuscript.

Thank you very much for pointing this out! We have now added the abbreviation to the revised manuscript as follows:

Recent estimates of this equilibrium climate sensitivity (ECS), using different lines of evidence, ...

Reviewer 4:

I was asked to review this resubmission in light of Reviewer 2's (R2's) concerns about the use of model constraints. The design of the study is ambitious and makes a number of assumptions to reach its conclusions. A main request of R2 was to make these assumptions more explicit throughout the paper. While the authors have clarified some important points and improved readability, I think that they have tried to argue away some sources of uncertainty that are in fact fundamental. Straightforward statements of the shortcomings of these assumptions are still needed following the suggestions of R2, including a discussion of implications for the accuracy of the confidence intervals given, which are an essential result. Specifically, numbered as in the rebuttal document:

In response to the concern about using Greenland temperature to represent global mean temperature, we now use the LOVECLIM simulated global mean temperature directly for the shorter period covering DO 5~12 in the main text. This does have a small impact on the estimated gains, but the values still provide a tighter constraint on the magnitude of the feedbacks and rule out existing high and low end estimates. We have preserved the original analysis based on using the Greenland temperatures and the longer time interval, but have moved this into the Supplementary.

1. The authors argue that "...it is sufficient that the model reproduces the spatial patterns of change reasonably well." I think it is currently somewhat incongruent within the paper to have an emphasis on uncertainty propagation paired with a less rigorous examination of a model used for a critical step. What does "reasonable" mean here? How might errors in the model or the assumption that the covariance of D-O

events looks like that of AMOC hosing in a low-resolution model translate into errors in the final estimate? While it's beyond the scope of the paper to quantify all of these possibilities, the authors must at least discuss them since it seems like a substantial shortcoming.

We are constrained to use model outputs because there is no reliable observational based estimate of global temperature. Although there are several model simulations that have mimicked D-O events, the LOVECLIM simulation is the only one that provides results for multiple D-Os with realistic orbital changes. We have examined the simulated patterns of change compared to available site-based reconstructions of the direction of temperature changes from Voelker et al. (2002) - the latest global compilation of observations - and now include a table quantifying these comparisons (Table S4) in addition to maps for each D-O event. These comparisons show that the broad-scale patterns of change for each of the D-Os is largely captured by the model. We fully recognise that no model is perfect, but it is difficult to quantify the magnitude of potential errors due to model biases in the absence of reliable reconstructions. Furthermore, while it is true that hosing experiments do not necessarily reproduce the structure or duration of D-O events, they do generally reproduce the speed and general magnitude of the warmings. We agree that it is important to discuss potential issues raised by using the model outputs, and the LOVECLIM model in particular, and we will expand the text in the Discussion to explain more clearly why we have done this and how it might impact our results, as follows:

We rely on the LOVECLIM model to derive estimates of global temperature because there are insufficient observationally based, quantitative reconstructions to estimate these reliably. Although a number of modelling groups have made simulations to mimic D-O events during the glacial by adding freshwater forcing (e.g. Zhang et al., 2014; Kawamura et al. 2017; Zhang et al., 2017; Zhang and Prange, 2020; Vettoretti et al., 2022) none of these have used realistic forcings for individual D-O events. Comparison of the spatial patterns of the LOVECLIM simulated temperature changes for individual D-O events with records from the Voelker (2002) data compilation (Supplementary Figs 2.1 ~ 2.8) indicate that there is good qualitative agreement in the sign of the change, with >75% of the grid cells being correctly predicted (Supplementary Table 4). Although LOVECLIM is a low-resolution model and the simulations were made with fixed cloud cover, neither of these constraints should have a major impact on the estimates of global temperature (Zhang et al., 2010). Furthermore, analyses based on estimating global temperature from observed temperature changes in Greenland over the interval between 80 and 20 ka using the relationship between simulated Greenland and global temperature obtained from the LOVECLIM simulations (see Supplementary Information) produce comparable estimates of feedback strength. Thus, although the use of model outputs is a potential source of additional uncertainty, in the absence of a compelling alternative this approach provides a way to estimate greenhouse-gas climate feedbacks on centennial scales.

We will add the following additional references

Kawamura, K., Abe-Ouchi, A., Motoyama, H., Ageta, Y., Aoki, S., Azuma, N., Fujii, Y., Fujita, K., Fujita, S., Fukui, K., et al.: State dependence of climatic instability over the past 720,000 years from Antarctic ice cores and climate modeling, *Science advances*, 3, e1600 446, 2017.

Vettoretti, G., Ditlevsen, P., Jochum, M., and Rasmussen, S. O.: Atmospheric CO2 control of spontaneous millennial-scale ice age climate oscillations, *Nature Geoscience*, pp. 1–7, 2022.

Zhang, X. and Prange, M.: Stability of the Atlantic overturning circulation under intermediate (MIS3) and full glacial (LGM) conditions and its relationship with Dansgaard-Oeschger climate variability, *Quaternary Science Reviews*, 242, 106–143, 2020.

Zhang, X., Lohmann, G., Knorr, G., and Purcell, C.: Abrupt glacial climate shifts controlled by ice sheet changes, *Nature*, 512, 290–294, 2014.

Zhang, X., Knorr, G., Lohmann, G., and Barker, S.: Abrupt North Atlantic circulation changes in response to gradual CO2 forcing in a glacial climate state, *Nature Geoscience*, 10, 518–523, 2017.

Zhang et al., 2010 is already in the reference list

Supplementary Table 4 is shown below:

Table S4. Count of the grid cells with the same warming/cooling trend between LOVECLIM simulated global mean temperature change and Voelker (2002) observed global mean temperature change for each D-O event.

D-O	Agreement number	Total number
5	50	58
6	49	58
7	48	58
8	49	58
9	44	58
10	52	58
11	53	58
12	48	58

2. Related to 1, the effects of model biases have not been quantified, so I don't think the authors can argue that they are small. "Bias" in this instance extend beyond a time-mean model bias to errors in regression coefficients, which can be large (see e.g. Amrhein et al. 2020 for a discussion of quantifying structural uncertainties arising from model regression relationships). Please discuss the implications of regression errors linking Greenland ice cores and global mean temperatures during D-O events for your results.

3. *The authors argue multiple times that a correlation observed over modern times between Greenland and global mean temperature points to a similar relationship for D-O events. However, this logic and its shortcomings needs to be explored more. Climate variability has more than one spatial mode, and it's not obvious why a regression coefficient computed over recent decades (in the presence of anthropogenic warming) should argue for a similar relationship for D-O events. Moreover, I find that the statement in the revised manuscript that "The resulting estimates of global mean temperature increase include a propagation of the uncertainties in this relationship" is not correct because it neglects these issues of model regression error.*

Comments 2 and 3 are no longer relevant since we now use LOVECLIM simulated global mean temperature directly, instead of using a regression coefficient to convert Greenland temperature to global mean temperature. However, we would point out that the estimates of gain obtained using this direct approach rather than the regression between Greenland and global mean temperature are not very different. We have preserved the old analysis in the Supplementary Information and refer to the similarity in the results in our Discussion as evidence for the robustness of the analysis (see text above).

4. *In summary, the authors need to lay out these sources of systematic error and their impacts on results clearly. In addition, I would strongly encourage the authors to provide an estimate of this uncertainty in their confidence intervals, which would strengthen the paper. One approach could be to take Greenland-global regression coefficients from LOVECLIM, use them to reconstruct global mean temperature from a Greenland location in another model, and compare that reconstruction to the true GMT value in that model. The resulting estimate of the effects of regression error could be propagated to the final results.*

It would be interesting to examine the relationship between Greenland temperature and global mean temperature during the D-O events, but it is not clear that this would be a meaningful test of the regression approach given that no other model has realistic time-varying forcings. What is clearly needed to improve estimates of centennial-scale GHG feedbacks is an expansion of the network of sites with quantitative temperature reconstructions for comparison with the Greenland temperature records - which is something the community is working on but is not yet available.

5. *Finally, I was also personally confused by the ability of the regression procedure to distinguish among feedbacks from the three GHGs considered. Given that the history of GHGs is not uncorrelated in time, how are feedbacks partitioned among the gases? Is the problem well-posed? It would help to show sensitivity to standard errors of radiative forcing (which can be important for partitioning solution variance in least-squares problems) and other relevant parameters to argue for robustness.*

We are able to make estimates of the gain associated with changes in the three GHGs separately because we use ice core observations of the observed changes in GHG concentration (see lines 76-77 in the revised manuscript). Note that the radiative forcing and the concentration are not the same thing. The radiative forcing of CO₂ is influenced by the concentration of N₂O, the radiative forcing of CH₄ is also influenced by the concentration of N₂O, and the radiative forcing of N₂O is influenced by CO₂ and CH₄.

However, their influences have been addressed in the equations to calculate radiative forcing (see equations 3.3 ~3.8). The radiative forcings are independent and additive (see IPCC AR6 chapter 5 & 7).

Equations 3.3 ~3.8 in our manuscript:

We calculated the radiative forcing³⁴ associated with the change between minimum and maximum values for each event, as follows:

CO₂:

$$\Delta R_C = (a_1(C - C_0)^2 + b_1|C - C_0| + c_1\bar{N} + 5.36) \times \ln\left(\frac{C}{C_0}\right) \quad (3.3)$$

$$\sigma_{\Delta R_C} = \sqrt{\left(\frac{\partial \Delta R_C}{\partial C}\right)^2 \sigma_C^2 + \left(\frac{\partial \Delta R_C}{\partial C_0}\right)^2 \sigma_{C_0}^2} \quad (3.4)$$

where $a_1 = -2.4 \times 10^{-7} \text{ W m}^{-2} \text{ ppm}^{-1}$, $b_1 = 7.2 \times 10^{-4} \text{ W m}^{-2} \text{ ppm}^{-1}$,
 $c_1 = -2.1 \times 10^{-4} \text{ W m}^{-2} \text{ ppb}^{-1}$

CH₄:

$$\Delta R_M = (a_2\bar{M} + b_2\bar{N} + 0.043) \times (\sqrt{M} - \sqrt{M_0}) \quad (3.5)$$

$$\sigma_{\Delta R_M} = \sqrt{\left(\frac{\partial \Delta R_M}{\partial M}\right)^2 \sigma_M^2 + \left(\frac{\partial \Delta R_M}{\partial M_0}\right)^2 \sigma_{M_0}^2} \quad (3.6)$$

where $a_2 = -1.3 \times 10^{-6} \text{ W m}^{-2} \text{ ppb}^{-1}$, $b_2 = -8.2 \times 10^{-6} \text{ W m}^{-2} \text{ ppb}^{-1}$

N₂O:

$$\Delta R_N = (a_3\bar{C} + b_3\bar{N} + c_3\bar{M} + 0.117) \times (\sqrt{N} - \sqrt{N_0}) \quad (3.7)$$

$$\sigma_{\Delta R_N} = \sqrt{\left(\frac{\partial \Delta R_N}{\partial N}\right)^2 \sigma_N^2 + \left(\frac{\partial \Delta R_N}{\partial N_0}\right)^2 \sigma_{N_0}^2} \quad (3.8)$$

where $a_3 = -8.0 \times 10^{-6} \text{ W m}^{-2} \text{ ppm}^{-1}$, $b_3 = 4.2 \times 10^{-6} \text{ W m}^{-2} \text{ ppb}^{-1}$,
 $c_3 = -4.9 \times 10^{-6} \text{ W m}^{-2} \text{ ppb}^{-1}$

C, M, N denote the maximum values identified for CO₂, CH₄ and N₂O, respectively; C₀, M₀, N₀ denote the minimum values identified for CO₂, CH₄ and N₂O, respectively; \bar{C} , \bar{M} , \bar{N} denote the mean values identified for CO₂, CH₄ and N₂O, respectively; ΔR_C , ΔR_M , ΔR_N denote the radiative forcing brought about by CO₂, CH₄ and N₂O, with their corresponding standard errors, $\sigma_{\Delta R_C}$, $\sigma_{\Delta R_M}$, $\sigma_{\Delta R_N}$, respectively.

7th Jul 22

Dear Ms Liu,

Please allow me to apologise again for the delay in sending a decision on your manuscript titled "Past rapid warmings as a constraint on greenhouse-gas climate feedbacks" has now been seen by our reviewer, whose comments appear below. In light of their advice I am delighted to say that we are happy, in principle, to publish a suitably revised version in Communications Earth & Environment under the open access CC BY license (Creative Commons Attribution v4.0 International License).

We therefore invite you to revise your paper one last time to address the remaining comments of our reviewer. At the same time we ask that you edit your manuscript to comply with our format requirements and to maximise the accessibility and therefore the impact of your work.

EDITORIAL REQUESTS:

Please review our specific editorial comments and requests regarding your manuscript in the attached "Editorial Requests Table". Please outline your response to each request in the right hand column. Please upload the completed table with your manuscript files.

SUBMISSION INFORMATION:

OPEN ACCESS:

Communications Earth & Environment is a fully open access journal. Articles are made freely accessible on publication under a [CC BY license](http://creativecommons.org/licenses/by/4.0) (Creative Commons Attribution 4.0 International License). This license allows maximum dissemination and re-use of open access materials and is preferred by many research funding bodies.

For further information about article processing charges, open access funding, and advice and support from Nature Research, please visit <https://www.nature.com/commsenv/article-processing-charges>

At acceptance, you will be provided with instructions for completing this CC BY license on behalf of all authors. This grants us the necessary permissions to publish your paper. Additionally, you will be asked to declare that all required third party permissions have been obtained, and to provide billing information in order to pay the article-processing charge (APC).

[link redacted]

Best regards,

Joe Aslin

Locum Chief Editor,
Communications Earth & Environment
<https://www.nature.com/commsenv/>
Twitter: @CommsEarth

REVIEWERS' COMMENTS:

Reviewer #4 (Remarks to the Author):

Liu et al. have addressed my previous concerns and I recommend the paper for publication.

I do have two minor comments. In the abstract, you indicate that GHG concentration changes are derived from ice core records; please add a similar attribution for temperature changes that they are derived from a forced low-resolution model. Second, in line 112, please say what is meant by "the x- and y-variables" in this context.

Response to reviewers

We thank the reviewers for their comments on our manuscript and provide a point-by-point response below.

Reviewer #4 (Remarks to the Author):

Liu et al. have addressed my previous concerns and I recommend the paper for publication. I do have two minor comments.

1. In the abstract, you indicate that GHG concentration changes are derived from ice core records; please add a similar attribution for temperature changes that they are derived from a forced low-resolution model.

Thank you very much for your comment ! We now added the text in blue in the abstract.

Here we use these events to quantify the centennial-scale feedback strength of CO₂, CH₄ and N₂O by relating global mean temperature changes, *simulated by an appropriately forced low-resolution climate model*, to the radiative forcing of these greenhouse gases derived from their concentration changes in ice-core records.

2. In line 112, please say what is meant by "the x- and y-variables" in this context.

Thank you very much for your comment ! We now added the text in blue in the manuscript.

A maximum likelihood method⁴² is used to derive this ratio because it considers uncertainty of both the x- and y-variables (*i.e. the driver and the response*), in contrast with ordinary least squares regression which assigns uncertainty only to the y-variable.